# Privacy Induces Robustness: Information-Computation Gaps and Sparse Mean Estimation

**Kristian Georgiev**
MIT EECS
Cambridge, MA 02139
`krisgrg@mit.edu`

**Samuel B. Hopkins**
MIT EECS
Cambridge, MA 02139
`samhop@mit.edu`

## Abstract

We establish a simple connection between robust and differentially-private algorithms: private mechanisms *which perform well with very high probability* are automatically robust in the sense that they retain accuracy even if a constant fraction of the samples they receive are adversarially corrupted. Since optimal mechanisms typically achieve these high success probabilities, our results imply that optimal private mechanisms for many basic statistics problems are robust.

We investigate the consequences of this observation for both algorithms and computational complexity across different statistical problems. Assuming the Brennan-Bresler secret-leakage planted clique conjecture, we demonstrate a fundamental tradeoff between computational efficiency, privacy leakage, and success probability for sparse mean estimation. Private algorithms which match this tradeoff are not yet known – we achieve that (up to polylogarithmic factors) in a polynomially-large range of parameters via the Sum-of-Squares method.

To establish an information-computation gap for private sparse mean estimation, we also design new (exponential-time) mechanisms using fewer samples than efficient algorithms must use. Finally, we give evidence for privacy-induced information-computation gaps for several other statistics and learning problems, including PAC learning parity functions and estimation of the mean of a multivariate Gaussian.

## 1   Introduction

Avoiding leakage of sensitive data and robustness to data corruption or model misspecification are often key goals for designers of statistical estimators. Both these properties admit mathematical formalizations, and a great deal of recent work in (algorithmic) statistics has gone into designing and analyzing algorithms that satisfy them.

*Differential privacy* (DP) is the gold-standard formal definition of privacy for algorithms processing sensitive data [DMNS06]. DP requires that the distribution of outputs of an algorithm (or "mechanism") $M$ is insensitive to exchanging a small number of individuals in its input dataset. It offers such strong guarantees against privacy leakage that in addition to its ongoing adoption in industry, the US Census Bureau employs DP to satisfy its legal mandate to protect privacy [AACM+22].

The $\eta$-*contamination model* is a stringent formalization of robustness against model misspecification. It generalizes the classical model of i.i.d. samples: a dataset is $\eta$-corrupted for some $\eta > 0$ if it is first drawn i.i.d, but then an $\eta$-fraction of samples have been arbitrarily corrupted by a malicious adversary (who may look at the whole dataset) [Hub65, Tuk75, Hub11]. An $\eta$-*robust* algorithm is one which maintains guarantees of accuracy when given $\eta$-corrupted samples.

36th Conference on Neural Information Processing Systems (NeurIPS 2022).

Both robustness and privacy demand that the output of some statistical method not "change too much" when one or a few input samples are modified arbitrarily. This conceptual similarity has not gone unnoticed: [DL09] observe that "robust statistical estimators present an excellent starting point for differentially private estimators," and recent works have even made good on this idea in high-dimensional settings, taking inspiration from robust statistics to design private mechanisms [LKKO21, HBK22]. However, a general account of which private algorithms can be made robust, or vice versa, remains an open problem. Here we take a small step by tackling:

*Question 1: When are private algorithms (also) robust?*

While common wisdom holds that privacy and robustness are not formally comparable, we give a meta-theorem Theorem 2.1 which quantifies the degree of robustness that private mechanisms exhibit: at a high level, mechanisms satisfying quantitatively-strong versions of DP are automatically robust.

This connection between robustness and privacy gives us the tools to investigate a second basic question, concerning tradeoffs among computational resources, privacy, and statistical accuracy. *Accuracy*-privacy tradeoffs appear in even the most elementary statistical settings. For instance, estimating the mean of a $d$-dimensional Gaussian to $\ell_2$ error $\alpha$ requires $\Theta(d/\alpha^2)$ samples non-privately, but subject to $\varepsilon$-DP* requires (roughly) $\Theta(d/\alpha^2 + d/(\alpha\varepsilon))$ samples [BKSW19]. While important in their own right, these two-way tradeoffs are not the whole story. In numerous cases, even including the Gaussian mean estimation problem, *computationally efficient algorithms which achieve the optimal privacy-accuracy tradeoffs are not known.* This brings up the question:

*Question 2: Does requiring differential privacy introduce computational barriers in statistics?*

While computational barriers to efficient private algorithms are known in some settings [GHRU13, Ull16, Bun20], these apply only to algorithms with *worst-case* accuracy guarantees – in statistical settings we are asking only for *average-case* accuracy guarantees (although privacy should still hold with respect to all possible datasets).

An archetypal problem for which computational barriers arise when estimators are required to satisfy criteria beyond accuracy is that of *sparse mean estimation*. The goal there is to estimate a $k$-sparse vector $\mu \in \mathbb{R}^d$ using independent samples from a distribution with mean $\mu$. With no requirements on privacy or robustness, this can be accomplished with $O(k \log d)$ samples in polynomial time via simple thresholding-based estimators. In exponential time, it is possible to retain $O(k \log d)$ sample complexity and satisfy privacy and robustness (as we show in this paper), but in polynomial time $\Omega(k^2)$ samples are required (under a variant of the planted clique conjecture) just to satisfy robustness [BB20]. Given the outlined connection between robustness and privacy, the existence of an information-computation gap for *robust* sparse mean estimation naturally leads to:

*Question 3: How many samples do poly-time private algorithms for sparse mean estimation require?*

## 1.1 Our Contributions

We make three main contributions: (a) a meta-theorem characterizing robustness of private mechanisms; (b) a case study of sparse mean estimation, including a computational lower bound arising from robustness, and a new Sum-of-Squares-based algorithm whose sample complexity matches that lower bound in parameter regimes where no such algorithms were previously known; and (c) a collection of computational and information-theoretic lower bounds for private mechanisms, inherited from lower bounds for robust algorithms.

**Robustness of Optimal Private Mechanisms**  Our first contribution is a simple but useful observation: *mechanisms with strong **group privacy** guarantees are automatically robust!* By "strong group privacy," we mean guarantees strong enough to retain privacy and accuracy when a constant fraction of individuals in the dataset are exchanged with others. We capture this in Theorem 2.1.

In spite of its simplicity, robustness of strongly-group-private mechanisms has significant consequences for simultaneously robust and private mechanisms, a topic of much recent interest [KMV21, LKO21, LKKO21, UKRK22, CS22], because mechanisms with optimal privacy-accuracy tradeoffs often automatically satisfy strong group privacy.

---

*See Section 2 for formal definitions.

Table 1: **Algorithms for sparse mean estimation.** In the input column, $\mathcal{N}$ signifies that the algorithm takes i.i.d. samples from $\mathcal{N}(\mu, I)$, and $\Sigma \preceq I$ — samples from a distribution with bounded covariance. The sample complexity column hides polylogarithmic factors in the ambient dimension $d$ and a priori mean bound (in $\ell_2$) $R$. In the auto-robust column we indicate whether our meta-theorem (Theorem 2.1) implies that the algorithm is robust to corruptions of an $\eta = 1/\text{poly} \log(d, R)$ fraction of the samples.

| Algorithm | Input | Runtime | Sample Complexity | Auto-robust |
|---|---|---|---|---|
| Hypothesis Sel. [BKSW19] | $\mathcal{N}$ | exp | $\frac{k+\log(1/\beta)}{\alpha\varepsilon} + \frac{k+\log(1/\beta)}{\alpha^2}$ | ✓ |
| Subset Sel. [Theorem C.7] | $\Sigma \preceq I$ | exp | $\frac{k+\log(1/\beta)}{\alpha^2\varepsilon}$ | ✓ |
| SoS [Theorem 1.2] | $\mathcal{N}$ | poly | $\frac{k^2+\log(1/\beta)}{\alpha^2\varepsilon}$ | ✓ |
| Threshold [Theorem 4.1] | $\mathcal{N}$ | poly | $\frac{k^2\log(1/\beta)}{\alpha^2\varepsilon}$ | ✗ |
| Peeling [CWZ21] [1] | $\mathcal{N}$ | poly | $R\left(\frac{k^{1.5}\log(1/\beta)}{\alpha\varepsilon} + \frac{k^{1.5}\log(1/\beta)}{\alpha^2}\right)$ | ✗ |

[1] Peeling is stated only for $(\varepsilon, \delta)$-DP in [CWZ21]. We use a (mildly) modified $\varepsilon$-DP version, which we formally state in Appendix G.

The privacy guarantees needed for automatic robustness rely on quantitatively-strong forms of differential privacy: either so-called *pure* DP, or *approximate* DP, but with the additive error parameter $\delta$ taken exponentially small, and high success probability. (By contrast, typical convention in the privacy literature is to take $\delta$ only polynomially small in other parameters.) For this reason, we focus primarily on mechanisms satisfying pure DP and achieving high success probability.

**Sparse Mean Estimation** Recall that in sparse mean estimation the goal is to estimate a $k$-sparse vector $\mu \in \mathbb{R}^d$ to $\ell_2$-error $\alpha$, succeeding with probability $1 - \beta$, using i.i.d. samples from a distribution with mean $\mu$. We contribute (a) new exponential-time $\varepsilon$-DP mechanisms using $O(k \log d)$ samples, (b) evidence that poly-time DP algorithms with high success probabilities require $\tilde{\Omega}(k^2)$ samples, and (c) new poly-time algorithms using $\tilde{O}(k^2)$ samples (in certain parameter regimes). One of these algorithms, using the SoS exponential mechanism of [HKM21], is our main technical contribution.

*Information-theoretic bounds:* First, as a baseline, we study sparse mean estimation without worrying about running time. We show that even subject to both privacy and robustness, $O(k \log d)$ samples suffice, with sample complexities differing in their dependence on $\alpha$ and $\varepsilon$ on between the cases that the underlying distribution is assumed to be Gaussian versus allowing for heavy-tailed distributions (assuming only bounded covariance), as in the case of non-sparse mean estimation. See the estimators Hypothesis Sel and Subset Sel in Table 1, and Appendix C for formal statements.

*Computational lower bound:* Next, we give evidence for a privacy-samples-success probability tradeoff for efficient private algorithms: a private algorithm for sparse mean estimation with high success probability will satisfy strong group privacy, and hence robustness, but efficient and robust algorithms for sparse mean estimation require $\Omega(k^2)$ samples (assuming the planted clique conjecture).

**Corollary 1.1** (of Theorem 2.1 and [BB20], Theorem 3.1). *Assume the* secret-leakage planted clique *conjecture [BB20]. For polynomially-related $n, k,$ and $d$, with $k = o(\sqrt{d})$, $\beta \in (0, 1)$, and $\alpha, \varepsilon > 0$, assume $\beta \leq 2^{-\varepsilon\sqrt{n}}$. Let $m(\beta, n, \varepsilon)$ be the greatest value less than $\log(1/\beta)/(\varepsilon n)$ in the set $\{n^{-o(1)}\} \cup \{n^{-1/(2t)} : t \in \mathbb{N}, t \geq 1\}$. Every polynomial-time, $\varepsilon$-DP algorithm which, for any $k$-sparse $\mu \in \mathbb{R}^d$ with $\|\mu\| \leq \text{poly}(d)$ can take $n$ samples from $\mathcal{N}(\mu, I)$ and return $\hat{\mu}$ such that $\|\hat{\mu} - \mu\| \leq \alpha$ with probability $1 - \beta$ requires $n \geq \frac{k^2 m^2}{\alpha^4 \cdot \text{poly} \log(d, \frac{1}{\varepsilon}, \frac{1}{\alpha})}$.*

Pretending $m = \log(1/\beta)/(\varepsilon n)$ and ignoring logarithmic factors in $d, \frac{1}{\varepsilon}, \frac{1}{\alpha}$, the lower bound says $n \gtrsim \left(\frac{k^2}{\alpha^2} \cdot \left(\frac{\log(1/\beta)}{\alpha\varepsilon}\right)^2\right)^{1/3}$ samples are required by efficient $\varepsilon$-DP algorithms for sparse mean estimation. This is the geometric mean of three terms: $\frac{k^2}{\alpha^2}$ and $\frac{\log(1/\beta)}{\alpha\varepsilon}$ (twice). We conjecture that the max of these is actually a lower bound: $n \gtrsim \frac{k^2}{\alpha^2} + \frac{\log(1/\beta)}{\alpha\varepsilon}$.

Even if this stronger lower bound were true, *existing efficient algorithms for sparse mean estimation would not match it*. The state-of-the-art for sparse $\varepsilon$-DP mean estimation using techniques in the literature is a modification (to achieve pure DP) of an algorithm by [CWZ21], whose sample complexity scales with $k^{1.5} \log(1/\beta)$ (see Appendix G).

*Efficient algorithms – SoS:* Are there polynomial-time algorithms which match the tradeoff from Corollary 1.1, or the stronger conjectured one above? Our main algorithmic contribution is a new algorithm for sparse mean estimation which matches the tradeoff of Corollary 1.1 (up to polylog factors) under the conditions: (1) $\alpha, \varepsilon \geq 1/\text{poly} \log(d)$, (2) $k^2 \approx \log(1/\beta)$, and (3) $k \geq d^{0.4}$. Of these conditions, (2) could be removed if the stronger conjectural lower bound above were true, while we believe that (1) and (3) are shortcomings of our algorithm, and might be removable. We are not aware of any previous efficient private algorithm which matches the above conjectured tradeoff for any simultaneously super-constant $k$ and $\log(1/\beta)$.

**Theorem 1.2.** *There exists $C > 0$ such that for every $\varepsilon, R > 0$, , $\alpha, \beta \in (0, 1)$, and large-enough $d, k \in \mathbb{N}$ such that $k \geq d^{0.4}$, there is a polynomial-time $\varepsilon$-DP algorithm $\mathtt{SoS}$ with the following guarantees. For every $k$-sparse $\mu \in \mathbb{R}^d$ with $\|\mu\| \leq R$, given $\eta$-corrupted samples $X_1, \ldots, X_n \sim \mathcal{N}(\mu, I)$, with probability at least $1 - \beta$, the algorithm outputs $\hat{\mu} \in \mathbb{R}^d$ such that $\|\mu - \hat{\mu}\| \leq \alpha + O\left(\sqrt{(\log(Rd))^C} \eta\right)$, so long as $n \gg (\log(Rd))^C \cdot \frac{k^2 + \log(1/\beta) + \log \log R}{\alpha^2 \varepsilon}$.*

Our algorithm employs the Sum-of-Squares exponential mechanism invented by [HKM21] for a private mean estimation algorithm, but adapting this approach to the sparse setting requires overcoming several technical roadblocks (see Section 3 and Appendix D). Our approach hits a technical obstacle related to the volume of the $d$-dimensional $\ell_1$ ball when $k \ll d^{0.4}$, and we leave as an open problem to match or approach the tradeoff in Corollary 1.1 for a wider range of parameters.

*Linear-time coordinate selection:* Finally, what can be accomplished with a "truly efficient" algorithm – one which does not require solving large semidefinite programs as in the SoS exponential mechanism? The state-of-the-art private algorithm for sparse mean estimation, of [CWZ21], is a simple iterative coordinate-selection procedure. With a minor modification to achieve a pure DP guarantee, that algorithm uses $O_R(k^{1.5} \log d)$ to estimate $k$-sparse mean vectors $\mu \in \mathbb{R}^d$ with $\|\mu\| \leq R$, with probability $1 - \beta$. However, the $O_R(\cdot)$ hides a *linear*, rather than logarithmic, dependence on $R$, which is very costly even for moderately-large values of $R$!

While it is now well understood how to obtain logarithmic-in-$R$ sample complexity for non-sparse mean estimation, standard approaches introduce a linear dependence on ambient dimension $d$. We improve over the algorithm of [CWZ21] while maintaining linear running time by designing a simple thresholding procedure for estimating the support of $\mu$ requiring a number of samples which is *independent of $R$*. Once the support is known, the ambient dimension of the problem can be reduced from $d$ to $k$, and a off-the-shelf private non-sparse mean estimation algorithm can be run. A formal statement is given in Section 4. We demonstrate with experiments on synthetic data (Section 3) that the coordinate-selection procedure $\mathtt{Threshold}$, is substantially more accurate than that used by prior state of the art [CWZ21].

By contrast to $\mathtt{SoS}$, the linear-time algorithm $\mathtt{Threshold}$ has sample complexity that scales with $k^2 \log(1/\beta)$ rather than $k^2 + \log(1/\beta)$. While this difference might appear minor at first, it has significant consequences: the linear-time algorithm cannot match the computational lower bound in Corollary 1.1, and it does not have strong-enough privacy guarantees to be robust via Theorem 2.1. Note that our results do not preclude other tradeoffs between $k$ and $\log(1/\beta)$. For instance, we leave it as an exciting open problem to design an efficient algorithm with sample complexity scaling with $k \log(1/\beta)$.

**Information-Computation Gaps in Private Statistics** A problem in private statistics has an *information-computation* gap if the accuracy-privacy tradeoffs achieved by optimal (exponential-time) mechanisms for that problem cannot be achieved by polynomial-time algorithms. Our connection between privacy and robustness can be used to give evidence for several information-computation gaps in private statistics beyond sparse mean estimation.

*Gaussian mean estimation* (Corollary B.2): Gaussian mean estimation, where the goal is to estimate $\mu \in \mathbb{R}^d$ given i.i.d. samples from $\mathcal{N}(\mu, I)$, is arguably even simpler than sparse mean estimation. We give evidence for an information-computation gap: polynomial-time $\varepsilon$-DP algorithms obtaining

accuracy $\alpha$ with probability at least $1 - \beta$ require $n \geq \log(1/\beta) \cdot (\log(1/\alpha))^{1/2-o(1)}/(\alpha\varepsilon))$ samples, unless there exist robust polynomial-time algorithms for Gaussian mean estimation which would contradict known statistical query lower bounds [DKS17] – this is a $\log(1/\alpha)^{1/2-o(1)}$ factor gap.

*Learning parities* (Corollary B.3): We also consider one of the most fundamental *supervised* learning problems: privately PAC learning parity functions from labeled examples. For each $S \subseteq [n]$, we can define a parity function $f_S : \{\pm 1\}^n \to \{\pm 1\}$ by $f_S(x) = \prod_{i \in S} x_i$. The goal is to take labeled examples $(x, y)$ drawn from some distribution $D$ and find a parity function $f_S$ such that $\Pr_{(x,y)\sim D}(f_S(x) = y) = 1$, assuming one exists. While polynomial-time private algorithms for learning parities are known [KLN+11], we show, via Theorem 2.1 that the failure probabilities of any such algorithms must be larger than what can be achieved in exponential time, or else $RP = NP$.

**Information-Theoretic Lower Bounds in Private Statistics** Finally, we show that the connection to robustness can provide information-theoretic lower bounds for private mechanisms. As an example, we study private covariance testing (Corollary B.4), where the goal is to take samples from $\mathcal{N}(0, \Sigma)$ and detect whether $\Sigma = I \in \mathbb{R}^{d \times d}$ or if $\|\Sigma - I\|_F \geq \gamma$. Appealing to the lower bound of [DK21] for robust covariance testing, we give a lower bound for private covariance testing, showing that $\Omega(d^2)$ samples are required by private algorithms with high success probabilities, while $O(d)$ suffice non-privately.

Formal statements for the results on Gaussian mean estimation, learning parities, and covariance testing can be found in Appendix B.

## 1.2 Related Work

*Privacy and robustness.* As mentioned in Section 1, there is a rich history of connections between DP and robustness, starting from the propose-test-release (PTR) framework of Dwork and Lei [DL09]. Building on top of PTR, a number of recent works tackle high-dimensional statistics problems by leveraging robust primitives [BGS+21, LKO21], themselves inspired by a recent revolution in high-dimensional robust statistics [DK19]. On the flip side, private algorithms for certain problems have been shown to "automatically" exhibit a small amount of robustness [TS13, HKM21].

*Sparse mean estimation.* Without privacy or robustness requirements, it is a folklore result that the truncated empirical mean achieves the information-theoretically optimal rate. In the approximate DP case, [TS13] show that the stability of LASSO can be leveraged for private support selection and private sparse regression via the sample-and-aggregate framework [NRS07]. Cai, Wang, and Zhang [CWZ21] show information-theoretic lower bounds for approximate DP (based on tracing attacks [HSR+08]) and computationally efficient algorithms that match those bounds with constant probability under additional assumptions on the $\ell_\infty$ norm of the mean $\mu$. In the presence of $\eta$-corruptions of the samples, [BDLS17] gives an $\tilde{O}\left(k^2 \log(d)/\eta^2\right)$-sample algorithm matching SQ lower bounds from [DKS17].

*Computational Roadblocks to Privacy.* Several prior works investigate computational roadblocks to privacy arising from cryptographic considerations, e.g. [Ull16, UV11]. The hard problem instances constructed in such works have a worst-case flavor, while we are interested in computational hardness for typical datasets/those drawn i.i.d. from an underlying probability distribution.

*Sum-of-Squares Method.* The SoS method for algorithm design in high-dimensional statistics has led to a number recent of algorithmic advances – see the survey [RSS18]. [HKM21], which introduces the SoS exponential mechanism, is most closely related, and provides the foundations for Theorem 1.2.

*Lower bounds for private mechanisms* There are multiple works leveraging group privacy to derive lower bounds for private algorithms, both for pure and approximate DP [HT10, De12, SU15, BS16]. They are all information-theoretic in nature, while we also provide computational hardness results. Additionally, to the best of our knowledge, there are no existing work relating lower bounds for privacy to ones for robustness.

## 2 Automatic Robustness Meta-Theorem and Private Robust Mechanisms

We formally define DP in Appendix A. With this in hand, With this in hand, we are ready to state and prove our meta-theorem on automatic robustness of private algorithms.

**Theorem 2.1** (Automatic Robustness Meta-Theorem). *Let $M : \mathcal{X}^* \to \mathcal{O}$ be an $(\varepsilon, \delta)$-private map from datasets $\mathcal{X}^*$ to outputs $\mathcal{O}$. For every dataset $X_1, \ldots, X_n$, let $G_{X_1,\ldots,X_n} \subseteq \mathcal{O}$ be a set of* good *outputs. Suppose that $M(X_1, \ldots, X_n) \in G_{X_1,\ldots,X_n}$ with probability at least $1 - \beta$ for some $\beta = \beta(n)$. Then, for every $n \in \mathbb{N}$, on $n$-element datasets $M$ is* robust *to adversarial corruption of any $\eta(n)$-fraction of inputs, where*

$$\eta(n) = O\left(\min\left(\frac{\log 1/\beta}{\varepsilon n}, \frac{\log 1/\delta}{\varepsilon n + \log n}\right)\right),$$

*meaning that for every $X_1, \ldots, X_n$ and $X'_1, \ldots, X'_n$ differing on only $\eta n$ elements, $M(X'_1, \ldots, X'_n) \in G_{X_1,\ldots,X_n}$ with probability at least $1 - \beta^{\Omega(1)}$.*

*Proof of Theorem 2.1.* Consider $\eta n$ intermediate datasets $(X_1, \ldots, X_n) = \mathbf{X}_0, \ldots, \mathbf{X}_{\eta n} = (X'_1, \ldots, X'_n)$, where a single coordinate $X_i$ is modified in passing from $\mathbf{X}_j$ to $\mathbf{X}_{j+1}$. Let $p_j = \Pr(M(\mathbf{X}_j) \in G_{X_1,\ldots,X_n})$. Then we have the following recurrence for $(1 - p_j)$:

$$(1 - p_j) \leq e^\varepsilon (1 - p_{j-1}) + \delta \text{ for } j \geq 1, \text{ and } (1 - p_0) \leq \beta,$$

from which we obtain $(1 - p_{\eta n}) \leq e^{\varepsilon \eta n}(\beta + \eta n \delta)$. The conclusion follows. $\qquad\square$

An analogous statement for concentrated DP [BS16] is presented in Appendix H.

Theorem 2.1 can be applied broadly to show that optimal private mechanisms are automatically robust, frequently even with optimal dependence of the lost accuracy on the rate of corruption. This breadth is possible because, for many statistical problems, *private mechanisms obtaining information-theoretically optimal privacy-accuracy tradeoffs automatically have the strong group privacy guarantees needed to apply Theorem 2.1.* This is because strong group privacy for a mechanism $M$ is implied by two other desirable properties of private mechanisms: (1) $M$ satisfies *pure* DP (or, $(\varepsilon, \delta)$-DP for small choices of $\delta$), and (2) $M$ produces accurate results *with high probability* over the randomness used internally by the mechanism. We give two examples of this phenomenon below.

*On the success probabilities of private algorithms:* Before turning to examples, we observe that Theorem 2.1 only gives robustness to a constant fraction of corrupted samples for private algorithms which have very high probability of succeeding – to obtain $\eta \geq \Omega(1)$ requires $\beta \leq 2^{-\Omega(n)}$. In most work on randomized algorithm design, the difference between succeeding with probability $2/3$ versus $1 - \beta$ for small $\beta$ can be treated as an afterthought, because algorithms can be repeated to amplify success probability. *But this kind of naive repetition causes privacy leakage!*

In spite of this, optimal private mechanisms in statistics often do succeed with high probability, using more sophisticated approaches than naive repetition: in fact, high success probability is generally implied by $M$'s outputs having (asymptotically) *optimal confidence intervals*. Our work points to a need for algorithm designers to focus on the confidence intervals/success probabilities of private algorithms: the payoff is robustness for free.

*On Black-Box Robustification of Optimal Private Mechanisms:* We observe that Theorem 2.1 can be used to automatically obtain robust and private mechanisms from private ones with high success probabilities. For instance, *(1) Hypothesis selection:* The private hypothesis selection procedure of [BKSW19], already proved robust against non-adaptive adversaries, is additionally robust to corruptions made by adaptive adversaries. *(2) Affine-invariant mean estimation:* [BGS+21, LKO21] study mean-estimation mechanisms which provide error guarantees in the *Mahalanobis distance* $\|\Sigma^{-1/2}(\hat\mu - \mu)\|$ given samples from $\mathcal{N}(\mu, \Sigma)$. [BGS+21] give a private mechanism for this problem with high success probability, and [LKO21] give a simultaneously private and robust mechanism. In either case, because both mechanisms provide strong-enough privacy guarantees to apply Theorem 2.1, robustness can be obtained in a black-box fashion knowing only the privacy guarantees.

*From Theorem 2.1 to Lower Bounds for Private Statistics:* As we discussed in Section 1.1, we use Theorem 2.1 to prove both computational and information-theoretic sample-complexity lower bounds for private algorithms for sparse mean estimation (Corollary 1.1), non-sparse mean estimation, learning parities, and covariance testing. We prove Corollary 1.1 in the next section, and defer the remaining statements and proofs of lower bounds to the supplement.

# 3 Sparse Private Mean Estimation: Techniques

In this section, we overview ideas which go into our results on sparse mean estimation, starting from the information-theoretic results, and moving on to computational barriers and polynomial-time algorithms.

**Information-theoretic bounds** In the absence of computational considerations, the landscape for *Gaussian* sparse mean estimation can be understood via standard tools: a packing-based lower bound, and a matching (exponential-time) mechanism can be constructed as a direct corollary of existing results in the literature on private hypothesis selection [BKSW19]. For completeness, we carry out those in Appendix C.

In the *heavy-tailed* case, assuming only that the samples $X_1, \ldots, X_n$ are drawn from a distribution with $k$-sparse mean $\mu$ and with bounded covariance, one can no longer construct a small cover of the set of possible distributions; this precludes an approach as general as hypothesis selection from directly applying. Instead, we design a mechanism which first selects a subset of $k$ coordinates, then hands off to a non-sparse mean estimation mechanism run just on those coordinates.

To select the coordinates, we use the exponential mechanism. To define a score function, we take inspiration from recent ideas in high-dimensional statistics [LM19] using empirical quantiles of univariate projections of the samples. The key idea for coordinate selection is to restrict attention to projection in sparse directions. For samples $X_1, \ldots, X_n$, we define the following score function on subsets of coordinates $T \subseteq [d]$:

$$S(\{X_i\}_{i \leq n}, T; L) = \max_{v \in \mathbb{R}^k, \|v\|_2 = 1} \sum_{i=1}^{n} \mathbb{1}\{v^\top X_{iT} \geq L\}, \tag{1}$$

parametrized by a scalar threshold $L^\dagger$. Then we sample $T$ from the distribution $\Pr(T) \propto \exp(\varepsilon S(\{X_i\}_{i \leq n}, T; L))$. Since $S$ has sensitivity $\Delta(S) = 1,^\ddagger$ the resulting mechanism satisfies $\varepsilon$-DP. We show using standard concentration tools that if $n \gg O(k \log(d) + \log(1/\beta)/(\alpha^2 \varepsilon))$ then this mechanism identifies a subset $T$ containing all but $\alpha$ of the $\ell_2$-mass of $\mu$, with probability at least $1 - \beta$. To do so, we choose the threshold $L$ such that with high probability all "bad" subsets containing a small portion of the $\ell_2$ mass of $\mu$ have "low" score compared to "good" ones for which $\|\mu_T\| \approx \|\mu\|$; finally, we use bucketing to control the variance of samples such that we can set $L$ as close to zero as possible without allowing "bad" subsets to achieve high score.

We then delegate the mean estimation on the candidate set of coordinates to any information-theoretically optimal (non-sparse) mean estimation mechanism. For the matching lower bound, standard packing-based arguments suffice (see Proposition C.13 for a formal argument).

**Computational barrier** Now we turn to the landscape when we require polynomial-time. Starting with barriers, we first show how to get Corollary 1.1 from our meta-theorem.

*Proof of Corollary 1.1.* First of all, since $n \geq \log(1/\beta)^2/\varepsilon^2$, there exists a value $m$ in the given set. Now suppose a polynomial-time private algorithm exists tolerating $n \leq k^2 m^2/(\alpha^4 \text{poly} \log(d, 1/\varepsilon, 1/\alpha))$. By adjusting parameters $\beta, \varepsilon$ to artificially weaken the guarantees as necessary, we may assume that $m(\beta, n, \varepsilon) = \log(1/\beta)/(\varepsilon n)$. Now, Theorem 3.1 of [BB20] shows that no algorithm for sparse mean estimation under these parameters can tolerate an $\eta$-fraction of adversarial corruptions with $k^2 \eta^2/\alpha^4 \gtrsim n$, where $\gtrsim$ hides polylogarithmic factors. But from Theorem 2.1 our hypothesized private algorithm tolerates an $\eta = \log(1/\beta)/(\varepsilon n)$ fraction of adversarial corruptions, which is a contradiction. □

**Overview of SoS Algorithm for Sparse Mean Estimation (Theorem 1.2)** We remain informal in this section and defer mathematical rigor to the supplement. For now, let $\alpha = \Theta(1)$. A standard trick reduces from $\alpha \ll 1$ to this case.

As with prior work on both private and robust mean estimation (e.g. [CFB19, HKM21]) our algorithm produces a series of iterates $x_0, x_1, \ldots, x_T$ for $T = O(\log d)$, where $x_0$ is the origin, with the invariant

---

$^\dagger$For a sample $X \in \mathbb{R}^d$, let $X_T$ to be the projection onto the basis vectors with indices in the set $T \subseteq [d]$.

$^\ddagger$That is, it changes by at most 1 when any sample is exchanged for another.

that $\|x_t - \mu\| \leq 0.9\|x_{t-1} - \mu\|$. To accomplish this, given $x_{t-1}$ with $\|\mu - x_{t-1}\| \gg 1$, and samples $X_1, \ldots, X_n \sim \mathcal{N}(\mu, I)$, we find a unit vector $v$ such that $\langle v, \mu - x_{t-1}\rangle \geq 0.9\|\mu - x_{t-1}\|$; then we could take $x_t = x_{t-1} + \Omega(\|\mu - x_{t-1}\|)v$.

Unlike prior works, for reasons we will see shortly, we also need the invariant that $x_t$ is $k$-sparse. So, we actually take $x_t$ to be $x_{t-1} + \Omega(\|\mu - x_{t-1}\|)v$ with all but the largest-magnitude $k$ coordinates set to 0; we show that this thresholding step cannot increase the distance to $\mu$ by too much (Lemma D.4).

**Picking a gradient in exponential time:** To find such a gradient vector $v$, we use the SoS exponential mechanism [HKM21]. Let us first see how we would pick a gradient vector $v$ using the (non-SoS) exponential mechanism, but allowing exponential running time. For a given dataset $\mathcal{X} = X_1, \ldots, X_n$, the goal is to find a *score function* $s_{\mathcal{X}}(v) \in \mathbb{R}$ which assigns each $(2k)$-*sparse* unit vector a score, such that for neighboring datasets $\mathcal{X}, \mathcal{X}'$ we always have $|s_{\mathcal{X}}(v) - s_{\mathcal{X}'}(v)| \leq 1$ ("bounded differences"). Here we choose $2k$-sparse because $\mu - x_{t-1}$ is itself $2k$ sparse. Then outputting a random $v$, where each is chosen with probability $\propto \exp(\varepsilon s_{\mathcal{X}}(v))$, gives an $\varepsilon$-DP mechanism. The goal is that $v$s with high scores are closer to $\mu - x_{t-1}$.

A good choice of score function turns out to be $s_{\mathcal{X}}(v) = \sum_{i \leq n} \mathbf{1}\{\langle v, X_i - x_{t-1}\rangle \geq \|x_{t-1} - \mu\| - O(1)\}$ – using standard concentration of measure one can show that as long as $n \gg k \log d$, for $v$ s.t. $s_{\mathcal{X}}(v) \geq 0.9n$ we will have $\langle v, \mu - x_{t-1}\rangle \geq 0.9\|\mu - x_{t-1}\|$ ("utility"). Furthermore, if one were to sample a uniformly random $k$-sparse unit vector, it would satisfy $\langle v, \mu - x_{t-1}\rangle \geq 0.9\|\mu - x_{t-1}\|$ with probability $d^{-O(k)}$ (since $\mu - x_{t-1}$ is $2k$-sparse). So, the distribution given by $\Pr(v) \propto \exp(\varepsilon s_{\mathcal{X}}(v))$ puts $1 - \beta$ probability on $v$ with $s_{\mathcal{X}}(v) \geq 0.9n$, so long as $n \gg (k \log d + \log(1/\beta))/\varepsilon$, since it "boosts" the probabilities of these high scoring vectors by a factor of $\exp(\Omega(d \log k)) \geq d^{\Omega(k)}$.

Here we have crucially used the fact that $x_{t-1}$, and hence $\mu - x_{t-1}$, is sparse: otherwise, the gradient $v$ we need to select would not be sparse, and we would need to use exponential mechanism to sample $v$ from a bigger set. This, in turn, would require us to draw more samples $n$, to ensure that the score function is well-behaved for a bigger set of vectors, because the probability of $\langle v, \mu - x_{t-1}\rangle \geq 0.9\|\mu - x_{t-1}\|$ for uniformly-random $v$ would be $\ll d^{-k}$.

Of course, the major drawback of the above is that it is not clear how to sample from the necessary distribution of $v$s efficiently – in fact, doing so would violate the lower bound of Corollary 1.1. There is also a second drawback: to evaluate the score function given above, we would need to know $\|\mu - x_{t-1}\|$; however, we are able to adapt the strategy of [HKM21] for this task to the sparse setting, re-using several of the ideas below.

**From exponential to polynomial time with SoS exponential mechanism:** The SoS exponential mechanism allows potentially exponential-time instances of the exponential mechanism to be converted into polynomial time algorithms, so long as (a) the bounded-differences and utility properties of the score function can be proved in a certain restricted proof system (the *SoS proof system*), and (b) the set over which the exponential mechanism is run is convex.

*Convexity:* The $2k$-sparse unit vectors – used by the above exponential-time algorithm – do not form a convex set. A natural idea is to relax from the $2k$-sparse vectors to the (scaled) $\ell_1$ ball. This creates a substantial difficulty: the set $\left\{v : \|v\|_1 \leq O\left(\sqrt{k}\right)\right\}$ has much more volume near the origin than the set of $2k$-sparse unit vectors. In particular, it is no longer true that $\langle v, \mu - x_{t-1}\rangle \geq 0.9\|\mu - x_{t-1}\|$ with probability $d^{-O(k)}$ for uniformly-random $v$; this probability will be exponentially small in $d$.

To fix this, we "fatten" the $\ell_1$ ball: we use the exponential mechanism over the set $C = \left\{v : \exists w \text{ s.t. } \|v - w\| \leq 0.01, \|w\| \leq 1, \|w\|_1 \leq O\left(\sqrt{k}\right)\right\}$. While perhaps counterintuitive that *adding* vectors is helpful here, it is possible (Lemma E.5) to show roughly the following statement: over randomly-chosen $v$ in this set, $\Pr(\langle v, \mu - x_{t-1}\rangle \geq 0.9\|\mu - x_{t-1}\|) \geq e^{-O(d/\sqrt{k})} \cdot d^{-O(k^2)} = d^{-O(k^2)}$ for $k \geq d^{0.4}$. To see this, note that all $w$ with $\|w - \frac{\mu - x_{t-1}}{\mu - x_{t-1}}\| \leq 0.01$ are in $C$ and have $\langle w, \mu - x_{t-1}\rangle \geq 0.0\|\mu - x_{t-1}\|$, and, using Sudakov minoration, $C$ can be covered by $d^{O(k^2)}$ $\ell_2$-balls of radius $1/\sqrt{k} + 0.01$. So at least a $d^{-O(k^2)} \left(\frac{0.01}{0.01 + 1/\sqrt{k}}\right)^d = d^{-O(k^2)} \exp(-O(d/\sqrt{k}))$-fraction of the volume of $C$ lies on such "good" $w$s. So, as long as $n \gg (k^2 \log d + \log(1/\beta))/\varepsilon$ and $k \geq d^{0.4}$, the arguments we used for the exponential-time method will still work.

*Utility and Bounded-Differences in SoS:* Finally, we turn to (a), the need to capture the proofs of the bounded-differences and utility in the SoS proof system. Here we take "intuitively simple" as a proxy for "expressible in SoS," deferring technical definitions to the supplement. The bounded-differences property for sum-of-threshold based score functions like $s_\chi$ above has a standard SoS proof [Hop20], so we won't address it further here.

The concentration of measure arguments we referred to above to establish utility are not captured by the SoS proof system! So we actually need to devise a new proof of utility for the score function $s_\chi$. The key step is a proof that if $n \gg (\log d)^{O(1)}(k^2 + \log(1/\beta))$ then with probability $1 - \beta$, for every unit vector $w$ with $\|w\|_1 \le \sqrt{k}$, we have $\sum_{i \le n} \mathbf{1}(\langle X_i - \mu, w \rangle \gg 1) \le 0.1n$. To see that this implies utility, note that if $v$ has $s_\chi(v) \ge 0.9n$, then there exists some $i$ such that $\langle v, X_i - x_{t-1} \rangle \ge \|\mu - x_{t-1}\| - O(1)$ but $\langle v, X_i - \mu \rangle \le O(1)$. Then $\langle v, \mu - x_{t-1} \rangle = \langle v, X_i - x_{t-1} \rangle + \langle v, \mu - X_i \rangle \ge \|\mu - x_{t-1}\| - O(1)$.

By a standard bounded-differences argument, it suffices to prove that $\mathbb{E} \max_w \sum_{i \le n} \mathbf{1}(\langle X_i - \mu, w \rangle \gg 1) \le 0.01n$. Letting $M = \sum_{i \le n}(X_i - \mu)(X_i - \mu)^\top$, by Cauchy-Schwarz, the quantity on the left-hand side is at most $0.01\sqrt{n} \cdot (\mathbb{E} \max_w \langle M, ww^\top \rangle)^{1/2}$. Splitting $M = M_{\text{diag}} + M_{\text{off-diag}}$, and applying Holder's inequality, $\langle M, ww^\top \rangle \le \|M_{\text{diag}}\|_\infty \|w\|_2^2 + \|M_{\text{off-diag}}\|_\infty \|w\|_1^2$, where $\| \cdot \|_\infty$ is the entry-wise $\ell_\infty$ norm. The important term here turns out to be the second one; it is possible to bound $\mathbb{E}\|M_{\text{off-diag}}\|_\infty \le (\log d)^{O(1)}\sqrt{n}$, for an overall bound of $0.01n^{3/4}k$ (up to log factors), which gives the desired bound if $n \gg k^2$. See Lemma E.3 and E.4 for the formal version of this argument.

## 4  Fast Algorithms and Experiments

In this section, we describe a practical algorithm — Algorithm 1 (Threshold in Table 1). We use standard DP tools to guarantee privacy and utility — the exponential mechanism [MT07] with a sensitivity-1 score function in a coordinate-wise fashion, followed by a black-box application of a univariate mean estimator.

---

**Algorithm 1** The subroutine exp−mech refers to the exponential mechanism [MT07], and KV−1D — to the univariate sparse mean estimator of [KV17].

---

**Input:** $\{x_i\}_{i \le n}, T, b, \sigma^2, n; \hat{\mu} \leftarrow \vec{0}$, selected $\leftarrow []$

1: $m_j \leftarrow \frac{1}{b} \sum_{j=i \cdot b}^{(i+1)b} x_i$ for $j \in [\lceil n/b \rceil]$      ▷ *compute bucketed means*
2: $z_i \leftarrow \sum_{j=1}^{\lceil n/b \rceil} \mathbb{1}\{(m_j)_i \ge T\}$ for $i \in [d]$.      ▷ *coordinate-wise threshold*
3: **for** $j \le k$ **do**
4:    $t \leftarrow$ exp-mech$\big(\{z_i\}_{i \in [d]\backslash\text{selected}}\big)$
5:    selected.insert($t$)
6:    $\hat{\mu}_t \leftarrow$ KV-1D$\big(\{(m_i)_t\}_{i=1}^d\big)$      ▷ *univariate estimation*

**Output:** mean estimate $\hat{\mu}$

---

**Theorem 4.1.** *Let* $X_1, \ldots, X_n \sim \mathcal{N}(\mu, \sigma^2)$. *Algorithm 1 is $\varepsilon$-DP outputs an estimate $\hat{\mu}$ s.t. $\|\hat{\mu} - \mu\|_2 \le \alpha$ with probability at least $1 - \beta$ as long as*

$$n = \Omega\left(\underbrace{\frac{k^2\sigma^2\left(\log d + \log(1/\beta)\right)}{\alpha^2\varepsilon}}_{\text{support estimation}} + \underbrace{\frac{\sigma^2 k \log(2k/\beta)}{\alpha^2} + \frac{\sigma k^{1.5}\log(2k/\beta)}{\alpha\varepsilon} + \frac{k\log(R)}{\varepsilon}}_{\text{dense estimation in } k \text{ dimensions}}\right) \tag{2}$$

$$= \Omega\left(\sigma^2\frac{k^2(\log d + \log(k/\beta))}{\alpha^2\varepsilon} + \frac{k\log(R)}{\varepsilon}\right).$$

The proof is given in Appendix F. While the statement and proof are given for Gaussians, we only use the concentration properties of the Gaussian distribution; hence the proof automatically applies for sub-Gaussian distributions as well.

Instead of the exponential mechanism, we could also use the peeling [DSZ18] algorithm, as is done for the linear-in-$R$ algorithm in [CWZ21].

## 4.1 Experimental details

Now we turn to empirically validating the performance of Algorithm 1. To isolate the effects of each subroutine (support estimation and dense mean estimation), we plot (1) the performance of the corresponding support estimation steps alone (2) $\ell_2$ error for both.

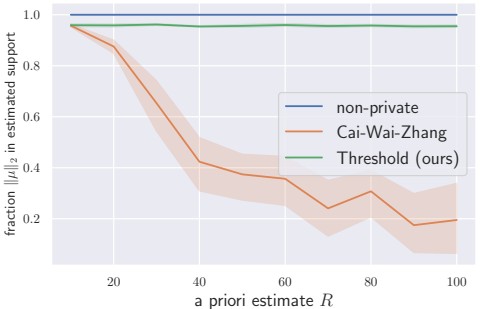

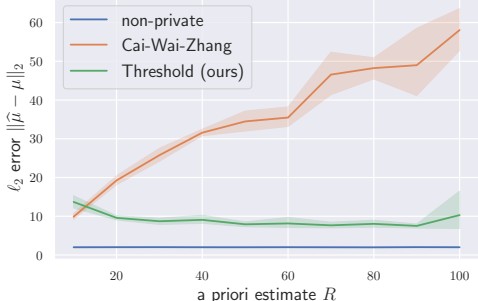

Figure 1: Empirical evaluation of 0.5-DP algorithms (and a non-private baseline) for support estimation for 1500 samples from $\mathcal{N}(\mu, I)$ in ambient dimension $d = 1000$ with $\|\mu\|_0 = 20$ (non-zero coordinates sampled uniformly from $[-10, 10]$) as a function of $R$, the a priori estimate of $\|\mu\|_2$.

Figure 2: Empirical evaluation of `Threshold` and the sparse mean estimation algorithm of [CWZ21] under $\varepsilon$-DP with $\varepsilon = 0.5$; results are shown for Gaussian data $X_1, \ldots, X_n \sim \mathcal{N}(\mu, 4 \cdot I)$ for a $k$-sparse $\mu$ in $\mathbb{R}^d$ for $k = 20, d = 1000, n = 1000$. The $\ell_2$ error of the estimates is plotted against the a priori mean estimate $R$. A non-private baseline is also presented to highlight the cost of privacy.

We address (1) in Fig. 1 — we use the fraction ($\ell_2$) mass of $\mu$ on the $k$ coordinates the algorithms select as a metric of success, since it is well-suited for mean estimation. Fig. 1 shows that (a) we significantly outperform the previous state of the art as soon as we introduce very mild uncertainty in the a priori estimate of $\|\mu\|$; (b) our method does not introduce additional constant factors "hiding" in the asymptotics.

In Fig. 1 we presented results for support estimation, in order to highlight the improvement coming from coordinate selection alone. In Fig. 2 we evaluate Algorithm 1 directly in terms of our metric of interest – $\ell_2$ error.

As can be seen from Fig. 2, the performance of the sparse mean estimation algorithm of [CWZ21] degrades rapidly even for very mild levels of uncertainty in the range of the mean - e.g. if we only know beforehand that the mean lies in the range $[-20, 20]$, instead of the tight range $[-10, 10]$, the $\ell_2$ error (expectedly) doubles when we use the [CWZ21] algorithm; in contrast, the performance of Algorithm 1 is effectively unchanged.

Finally, we highlight a weakness of Algorithm 1 – we gain the mild dependence on $R$ at the cost of losing the $\left(\frac{1}{\alpha\varepsilon} + \frac{1}{\alpha^2}\right)$-like sub-Gaussian rate; instead we have heavy-tailed-style $\left(\frac{1}{\alpha^2\varepsilon}\right)$-like rate. This comes with a practical cost in scenarios where the standard deviation $\sigma$ is much larger than the a priori mean estimate $R$.

For all figures, we average results over 10 random seeds and report average results, together with 95% bootstrap confidence intervals. Code necessary to reproduce all experiments is available[§]. For all experiments we use commodity hardware (CPU: Intel Core i7-9750H CPU @ 2.60GHz).

## 5 Acknowledgements

Work supported in part by the NSF grants CCF-1553428 and CNS-1815221. This material is based upon work supported by the Defense Advanced Research Projects Agency (DARPA) under Contract No. HR001120C0015.

---

[§]https://github.com/kristian-georgiev/privacy-induces-robustness

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
