# A  Preliminaries

We first introduce central definitions and results from differential privacy that we use throughout the paper. Next, we give background on the notion of robustness we work with. Finally, we briefly overview the central ideas in the Sum-of-Squares (SoS) tools we use.

**Differential Privacy**  We start by formally defining the notion of differential privacy.

**Definition A.1** ((Approximate) Differential Privacy). *Let $\mathcal{X}$ be a set and $\mathcal{X}^* = \{(X_1, \dots, X_n) : n \in \mathbb{N}, X_i \in \mathcal{X}\}$ be all possible datasets over $\mathcal{X}$. For $\varepsilon, \delta > 0$, a (randomized) map $M : \mathcal{X}^* \to \mathcal{O}$ (where $\mathcal{O}$ is an* output *set) is $(\varepsilon, \delta)$-DP if for every $(X_1, \dots, X_n), (X_1', \dots, X_n') \in \mathcal{X}^*$ such that $X_i = X_i'$ except for a single index $i$ and for every subset $S \subseteq \mathcal{O}$, $\Pr(M(X_1, \dots, X_n) \in S) \leq e^\varepsilon \Pr(M(X_1', \dots, X_n') \in S) + \delta$.*

The special case of $\delta = 0$ is referred to as *pure* DP and will be the main focus of our work. Throughout the paper, we refer to datasets that differ in one entry as *neighboring*.

We freely use basic primitives in private algorithm design like composition of private mechanisms, which we state for completeness.

**Lemma A.2** ((Basic) Composition [DMNS06]). *Suppose that we have private mechanisms $M_1, \dots, M_n$ where $M_i$ is $\varepsilon_i$-DP. Then an adaptive composition $M$ of $\{M_1, \dots, M_n\}$ is $\sum_{i \leq n} \varepsilon_i$-DP.*

We also freely use basic DP mechanism like the Laplace mechanism defined below.

**Lemma A.3** (Laplace mechanism [DMNS06]). *Let $f : \mathcal{X}^n \to \mathbb{R}^d$ such that $\max_{X, X' \text{ neighbors}} \|f(X) - f(X')\|_1 = \Delta$. Then the mechanism $M : \mathcal{X}^n \to \mathbb{R}^d$ defined as $M(X) = f(X) + L$, where $L \sim Lap\left(0, \frac{\varepsilon}{\Delta}\right)$ is sampled from the Laplace distribution, is $\varepsilon$-DP.*

The quantity $\Delta$ in Lemma A.3 is referred to as the $\ell_1$ sensitivity of $f$. A central part of our design of private mean estimation algorithms revolves around using functions with low sensitivity as primitives for our private estimators.

We make extensive use of the exponential mechanism [MT07]. It is a technique to privately select the (approximately) "best" object in a universe $\mathcal{H}$ according to a *score function $S$*, which measures the "goodness" of a given object. An important property of the score function is its *sensitivity $\Delta_S$*, defined as $\max_{h \in \mathcal{H}, \text{neighboring } D, D'} |S(D, h) - S(D', h)|$. Given a score function $S$ with sensitivity $\Delta_S$ and a privacy parameter $\varepsilon$, the exponential mechanism samples an object $h$ with probability proportional to $\exp\left(\varepsilon/(2\Delta_S) \cdot S(h)\right)$. The exponential mechanism comes with the following privacy and utility guarantees.

**Theorem A.4** ([MT07]). *For a dataset $X$ and a score function $S : \mathcal{X}^n \times \mathcal{H} \to \mathbb{R}$, the exponential mechanism $M$ on the score function $S$ is $\varepsilon$-DP and with probability at least $1 - \beta$ outputs an object such that*

$$S(M(X)) \geq OPT(X) - \frac{2\Delta_S}{\varepsilon}\left(\log\left(\frac{|\mathcal{H}|}{|\mathcal{H}^*|} + \log(1/\beta)\right)\right),$$

*where $\mathcal{H}^*$ is the set of objects achieving score $OPT$.*

**Robustness**  There are a variety of models for the adversary in the agnostic setting. We work with the case of *adaptive* adversaries:

**Definition A.5** ($\eta$-contamination model). *In the $\eta$-contamination model, given a "clean" distribution $D$, to draw $n$ ($\eta$-contaminated) samples from $D$, first draw $X_1', \dots, X_n' \sim D$, and then output any $\{X_1, \dots, X_n\}$ such that $X_i = X_i'$ for at least $(1 - \eta)n$ choices of $i$.*

Another popular model in the literature is that of an *oblivious* adversary who provides a distribution that is close in total variation distance to the original one. The name comes from the fact that, unlike in Definition A.5, the adversary is not allowed to inspect the samples and adaptively decide what samples to add and remove.

**Sum of Squares**  In Appendix D we use the SoS exponential mechanism in a black-box fashion. For a derivation of it, as well as examples, see [HKM21]. Informally, we use SoS as a proofs-to-algorithms paradigm which automatically "simple" (expressible within the SoS proof system) proofs into polynomial time algorithms. For an overview of SoS, see e.g. [BS14].

# B  Lower Bounds in Private Statistics

In this section, we state and prove theorems giving information-computation gaps for private algorithms for Gaussian mean estimation and learning parities, and we give an information-theoretic lower bound for private covariance testing, again via Theorem 2.1.

## B.1  Private Gaussian Mean Estimation

Consider the task of privately estimating the mean of class of spherical Gaussian distributions, and the following conjecture, supported by statistical query lower bounds [DKS17].

**Conjecture B.1.** *There exists $c > 0$, no $\mathrm{poly}(n, d, 1/\eta)$-time algorithm $\eta$-robustly estimates the mean of an unknown $d$-dimensional spherical Gaussian $\mathcal{N}(\mu, I)$ with, $\|\mu\| \leq \mathrm{poly}(d)$, from independent samples to $\ell_2$ error $\eta(\log 1/\eta)^{1/2-\Omega(1)}$, so long as $\eta \geq 2^{-d^c}$.*

(By contrast, $O(\eta)$ error is achievable in exponential time.) We prove the following corollary:

**Corollary B.2.** *Assume Conjecture B.1. Every $\mathrm{poly}(n, d, 1/\varepsilon)$-time $\varepsilon$-DP algorithm which takes i.i.d. samples from $\mathcal{N}(\mu, I)$ for $\mu \in \mathbb{R}^d$ and outputs a vector $\hat{\mu}$ such that $\|\hat{\mu} - \mu\| \leq \alpha$ with probability $1 - \beta$ for all $\|\mu\| \leq \mathrm{poly}(d)$ requires $n \geq \min\{\frac{\log(1/\beta)}{\varepsilon\alpha} \cdot (\log(1/\alpha))^{1/2-o(1)}, \frac{2^{d^c} \log(1/\beta)}{\varepsilon}\}$, where $c > 0$ is a universal constant.*

To see that Corollary B.2 captures an information-computation gap, recall that in exponential time it is possible to perform $\varepsilon$-DP Gaussian mean estimation for $\|\mu\| \leq \mathrm{poly}(d)$ with

$$n = \Theta\left(\frac{d + \log(1/\beta)}{\alpha^2} + \frac{d + \log(1/\beta)}{\alpha\varepsilon} + \frac{d\log d + \log(1/\beta)}{\varepsilon}\right)$$

samples [BKSW19]. Unless the SQ lower bounds of [DKS17] are broken by some polynomial-time algorithm, this error rate can be matched, at best, up to logarithmic factors by polynomial time algorithms.

*Proof of Corollary B.2.* A polynomial-time private algorithm with the hypothesized guarantees would be $\eta = \log(1/\beta)/(\varepsilon n)$-robust by Theorem 2.1. So long as $\log(1/\beta)/\varepsilon n = \eta \geq 2^{-d^c}$ this satisfies the hypotheses of Conjecture B.1, so it can't estimate $\mu$ to error $\eta(\log(1/\eta))^{1/2-o(1)}$; the conclusion follows by substituting $\log(1/\beta)/(\varepsilon n)$ for $\eta$. $\qquad\square$

## B.2  Privately Learning Parities

We now turn to a fundamental supervised problem - PAC learning parities. In the absence of noise, Gaussian elimination provides a polynomial-time sample-efficient algorithm for learning parities.

Learning under adversarial label noise is known as agnostic learning [KSS94]. Learning parities in the agnostic case, unlike in the noiseless case, is notoriously difficult; in the proper case, the problem has been shown to be NP-hard [Hås01], even for getting accuracy of $1/2 + \varepsilon$ for any $\varepsilon > 0$. With this in mind, our goal is to probe where does private parity learning lie computationally.

More formally, let PARITY be the class of functions $f_S : \{\pm 1\}^d \to \{\pm 1\}$ defined by $f_S(x) = \prod_{i \in S} x_i$ for each $S \subseteq [n]$. An algorithm PAC learns PARITY if, for every $S$, given $n$ samples $(x, f_S(x))$ where $x$ comes from some distribution $D$ on $\{\pm 1\}^d$, it finds $T$ such that $\Pr_{x \sim D}(f_S(x) \neq f_T(x)) \leq \alpha$. [KLN+11] show that

$$n = O\left(\frac{d\log(1/\beta)}{\varepsilon\alpha}\right) \tag{3}$$

suffice to privately PAC learn PARITY in polynomial time [KLN+11, Theorem 4.4], succeeding with probability $1 - \beta$. In contrast, allowing exponential time, only

$$n = O\left(\frac{d + \log(1/\beta)}{\varepsilon\alpha}\right) \tag{4}$$

suffice. From Theorem 2.1 we obtain the result that this gap cannot be closed with polynomial-time private algorithms, unless $RP = NP$:

**Corollary B.3.** *Suppose $RP \neq NP$. Then every polynomial-time $\varepsilon$-DP algorithm which for any $\beta > 0$ can PAC-learn $d$-variable* `PARITYs` *to accuracy $\alpha$, succeeding with probability $1 - \beta$, requires $n \geq \omega\left(\frac{\log(1/\beta)}{\varepsilon\alpha}\right)$ samples.*

*Proof.* Suppose otherwise, that there exists a private PAC learner using $n \leq O(\log(1/\beta)/(\alpha\varepsilon))$ samples. By Theorem 2.1, such an algorithm is also $\eta = \frac{\log(1/\beta)}{\varepsilon n}$-robust, with $\eta \geq \Omega(\alpha)$. We claim that it can be used to distinguish $1 - \Omega(\alpha)$-satisfiable instances of XOR-SAT from $1/2 + O(\alpha)$-satisfiable ones; by [Hås01] this is NP-hard for every constant $\alpha > 0$.

Given a $d$-variable instance $\phi$ of XOR-SAT, let $D$ be the following distribution on the hypercube. First, draw a clause $\prod_{i \in C} y_i = b_C$ from $\phi$ uniformly at random. Then let $c \in \{\pm 1\}^d$ be given by $c_i = -1$ if $i \in C$ and otherwise $c_i = -1$.

Suppose $\phi$ is $1 - \alpha'$-satisfiable, by some $x \in \{\pm 1\}^n$. Define $f_x(c) = \prod_{i\,:\,x_i=-1} c_i$. We claim that $\Pr_{C \sim \phi}(f_x(c) = b_C) \geq 1 - \alpha'$. This holds because $f_x(c) = \prod_{i\,:\,x_i=-1} c_i = \prod_{i\,:\,c_i=-1} x_i = b_C$ if $C$ is a clause satisfied by $x$, and by hypothesis a $(1 - \alpha')$-fraction of the clauses in $\phi$ are satisfied by $x$.

Now, given samples $(c, f_x(c))$, our hypothesized private PAC learner returns, with probability $1 - \beta$, some parity function $f_S$ such that $\Pr_{C \sim \phi}(f_x(c) = f_S(c)) \geq 1 - \alpha'$. This means that $\Pr_{C \sim \phi}(f_S(c) = b_C) \geq 1 - 2\alpha'$.

By $\eta$-robustness with $\eta \geq \Omega(\alpha)$ for $\alpha' \ll \alpha$, even given samples $(c, b_C)$, the learning algorithm returns such an $f_S$ with probability at least 0.99, taking $\beta$ sufficiently small. But given $f_S$ it is easy to extract a $1 - O(\alpha')$-satisfying assignment to $\phi$, which is NP hard. $\square$

## B.3 Private Covariance Testing

A fundamental question in high-dimensional statistics is that of distributional property testing [BFR$^+$00]. Given independent samples, the goal is to design efficient algorithms to test whether their distribution satisfies a given property.

In this work, we are interested in a simple problem in that field - Gaussian covariance testing. In particular, given samples $X_1, \ldots, X_n$ in $\mathbb{R}^d$ from $\mathcal{N}(0, \Sigma)$ with unknown $\Sigma$, we want to *privately* determine whether $\Sigma = I$ or $\|\Sigma - I\|_F \geq \gamma$ with high probability using as few samples as possible.

In the non-private case, [CM13] show that Gaussian covariance testing can be achieved with $O\left(\frac{d}{\gamma^2}\right)$ samples, significantly less than the $O\left(\frac{d^2}{\gamma^2}\right)$ samples required to learn the distribution. Curiously, Diakonikolas and Kane [DK21] show that under $\eta$-corruptions for any constant $\eta > 0$ one needs $\Omega(d^2)$ samples for testing.

[ADK$^+$19] show that one needs $O\left(\frac{d^2}{\gamma^2\varepsilon}\right)$ samples to privately learn the covariance of a Gaussian in $d$ dimensions. Thus, this is a natural upper bound on the sample complexity of *testing* the covariance.

We leverage Theorem 2.1 to show that any private algorithm with strong group privacy guarantees requires $\Omega(d^2)$ samples for covariance testing.

**Corollary B.4.** *For every $C > 0$ there is $C' > 0$ such that every $\varepsilon$-DP mechanism which takes independent samples $X_1, \ldots, X_n$ from $\mathcal{N}(0, \Sigma)$ for an unknown $\Sigma$ and distinguishes with probability at least $1 - \beta$ whether $\Sigma = I$ or $\|\Sigma - I\|_F \geq 1/2$ takes $n \geq \Omega(\min(C \log(1/\beta)/\varepsilon, C'd^2))$ samples.*

*Proof.* [DK21] shows that for every $c > 0$ there is $c' > 0$ such that any $c$-robust algorithm for distinguishing $\Sigma = I$ from $\|\Sigma - I\| > 1/2$ requires at least $c'd^2$ samples. Any $\varepsilon$-private algorithm which accomplishes this task with probability at least $1 - \beta$ will be $\log(1/\beta)/(\varepsilon n)$-robust, by Theorem 2.1. If $n \leq C \log(1/\beta)/\varepsilon$, then this algorithm is $\Omega(1/C)$-robust. Hence, there is $C'$ such that it must use $n \geq C'd^2$ samples. $\square$

# C  Information-theoretic Results for Private Sparse Mean Estimation

In this section we focus on algorithms and lower bounds for private sparse mean estimation in the absence of computational considerations. First, we consider the Gaussian case; then we develop algorithms for the more general heavy-tailed case where we only assume that the data has bounded covariance.

## C.1  Gaussian Private Sparse Mean Estimation

**Proposition C.1.** *For every $\alpha, \varepsilon, \beta, R > 0$ and small-enough $\eta > 0$ there is an $\varepsilon$-DP, mechanism taking $n \gg \frac{k \log(d)}{\alpha^2} + \frac{\log(1/\beta) + k \log(Rk/\alpha)}{\alpha \varepsilon}$ $\eta$-corrupted samples from $\mathcal{N}(\mu, I)$ with $\mu \in \mathbb{R}^d$, $\|\mu\|_0 \leq k$, and $\|\mu\| \leq R$ and produces $\hat{\mu}$ such that $\|\mu - \hat{\mu}\| \leq \alpha + O(\eta)$, with probability $1 - \beta$.*

Proposition C.1 follows quickly from combining the private hypothesis selection mechanism of [BKSW19] with Theorem 2.1 to establish robustness. For convenience, we restate the result of [BKSW19]. Before that, we introduce the notion of a Scheffé set.

**Definition C.2** (Scheffé set). *Let $\mathcal{H}$ be a set of distributions on $\mathcal{X}$. The Scheffé set for $H, H' \in \mathcal{H}$ is defined as $w(H, H') = \{x \in \mathcal{X} \mid H(x) > H'(x)\}$. The set $\mathcal{W}$ of Scheffé sets for $\mathcal{H}$ is given by $\mathcal{W}(\mathcal{H}) = \{w(H, H') \mid H, H' \in \mathcal{H}\}$.*

**Theorem C.3** ([BKSW19, Theorem 4.1]). *Suppose $\mathcal{H}$ is a set of distributions on $\mathcal{X}$. Let $d$ be the VC dimension of the set of indicators of the Scheffé sets of $\mathcal{H}$. Then there exists an $(\varepsilon, \delta)$-differentially private mechanism with the following guarantees. Suppose $D = \{X_1, \ldots, X_n\}$ is a set of private samples independently drawn from an unknown distribution $P$ and suppose there exists some $H^*$ such that $d_{\mathrm{TV}}(P, H^*) \leq \alpha$. If $n = \Omega\left(\frac{d + \log(1/\beta)}{\alpha^2} + \frac{\log(|\mathcal{H}|/\beta) + \min\{\log(|\mathcal{H}|), \log(1/\delta)\}}{\alpha \varepsilon}\right)$, then the output $\hat{H}$ of the algorithm is such that $d_{\mathrm{TV}}(P, \hat{H}) \leq 7\alpha$ with probability at least $1 - \beta$.*

Intuitively, Theorem C.3 tells us that if we have a hypothesis class of candidate distributions with a small cover, an example of which are Gaussians with bounded mean and known covariance, then we can privately select a distribution close to an unknown target distribution in total variation distance with only a mild additional cost of privacy. To make this more precise, we need the following definition.

**Definition C.4.** *A set $C_\alpha$ of distributions is an $\alpha$-cover for a set of distributions $\mathcal{H}$ if for every $H \in \mathcal{H}$, there is some $C \in C_\alpha$ such that $d_{\mathrm{TV}}(H, C) \leq \alpha$.*

In order to apply Theorem C.3 to our use case, we first show a cover for Gaussian distribution with $k$-sparse bounded means.

**Lemma C.5.** *Let $\mathcal{S}$ be the set of Gaussian distributions $\mathcal{N}(\mu, I)$ in $d$ dimensions with $k$-sparse $\mu$ such that $\|\mu\|_2 \leq R$. Then $\mathcal{S}$ admits an $\alpha$-cover of size*

$$O\left(\binom{d}{k} \left(\frac{Rk}{\alpha}\right)^k\right).$$

*Proof.* First, observe that it suffices to obtain an $\alpha$-cover for each of the $\binom{d}{k}$ possible choices of the support of $\mu$, and return the union of the covers. For any particular subset, the problem becomes equivalent to finding an $\alpha$-cover of the set of non-sparse Gaussian distributions with bounded mean in $k$ dimensions. By Lemma 6.8 in [BKSW19], there exists an $\alpha$-cover of size $O\left(\frac{Rk}{\alpha}\right)$. For completeness, we note that the proof of Lemma 6.8 in [BKSW19] constructs the $\alpha$-cover by taking the Cartesian product of $(\alpha/k)$-covers in each standard basis direction, which by the triangle inequality gives an $\alpha$-cover. $\square$

Covering numbers are tightly related to VC dimension. The final piece we need for Proposition C.1 is a bound on the VC dimension of the Scheffé sets of the set of $k$-sparse $d$-dimensional Gaussians with identity covariance.

**Lemma C.6.** *Let $\mathcal{H}_k^d$ be the set of Gaussian distributions with covariance matrix $I_d$ and mean $\mu \in \mathbb{R}^d$ such that $\|\mu\|_0 \leq k$. Let $\mathcal{W}\left(\mathcal{H}_k^d\right)$ be the set of indicators of Scheffé sets of $\mathcal{H}_k^d$. Then $\mathcal{W}$ has VC dimension at most $4k \log(de) = O(k \log(d))$.*

*Proof.* We want to bound the VC dimension of the set of functions $f_{H,H'} : \mathcal{X} \to \{0,1\}$ defined by $f_{H,H'}(x) \mapsto \mathbb{1}\{H(x) > H'(x)\}$ for $H, H' \in \mathcal{H}_k^d$.

First, observe that for any two Gaussians $H, H'$, $f_{H,H'}$ correspond to a linear threshold function $L(x) = \left\{a \in \mathbb{R}^d \mid \langle a, x \rangle \geq 0\right\}$. Restricting $H, H'$ to belong to $\mathcal{H}_k^d$, i.e. have $k$-sparse means, we now have that each $f_{H,H'}$ corresponds to a linear threshold function $L(x)$ generated by $2k$-sparse vector $x$. We now leverage a result of Ahsen and Vidyasagar [AV19, Theorem 6] which states that the set of linear threshold functions generated by $k$-sparse vectors in $d$ dimensions has VC dimension at most $2k \log(de)$; this concludes the argument. $\square$

We are now ready to prove Proposition C.1.

*Proof of Proposition C.1.* Consider the case of uncorrupted data ($\eta = 0$). We construct an estimator that gives a guarantee in $\ell_2$ distance, which we show is equivalent to total variation distance for spherical Gaussians.

From Lemma C.5 and a bound on the binomial coefficients we obtain a cover of size

$$O\left((de/k)^k \left(\frac{Rk}{\alpha}\right)^k\right). \tag{5}$$

From Lemma C.6, we know that the VC dimension of the set of indicators of the Scheffé sets of $k$-sparse Gaussians with identity covariance in $d$ dimensions is $O(k \log(d))$.

With the size of the $\alpha$-cover and the bound on the VC dimension established, we are ready to invoke Theorem C.3. This gives us the guarantee that for the output distribution $\hat{H}$ we have $d_{\mathrm{TV}}(P, \hat{H}) \leq 7\alpha$ with probability at least $1 - \beta$. Suppose the selected distribution $\hat{H}$ has mean $\hat{\mu}$. Since both $P$ and $\hat{H}$ are spherical Gaussians, we have that

$$\|\mu - \hat{\mu}\|_2 \leq d_{\mathrm{TV}}(P, \hat{H}) \tag{6}$$

following from [DMR18, Theorem 1.2]. Therefore, we can obtain an estimate with the desired properties by outputting the mean of the selected distribution $\hat{H}$.

Finally, robustness follows directly from Theorem 2.1. $\square$

As for an information-theoretic lower bound, [CWZ21] give a tracing-based lower bound (which also holds for approximate DP) which states that any $\varepsilon$-DP algorithm must use at least $n = \Omega\left(\frac{k \log(d)}{\alpha^2} + \frac{k \log(d)}{\alpha \varepsilon}\right)$ samples to estimate the mean of a sub-Gaussian distribution up to $\ell_2$ error $\alpha$, assuming $\|\mu\|_\infty \leq 1$. Alternatively, we can leverage hypothesis-selection based arguments again, to get a $\Omega\left(\frac{k \log(d) + \log(Rk/\alpha)}{\varepsilon}\right)$ bound, assuming $\|\mu\|_2 \leq R$.

## C.2 Heavy-Tailed Private Sparse Mean Estimation

Now we turn to a more general set of distributions – ones with bounded covariance. We give an (exponential-time) algorithm – Algorithm 2, which has strong group privacy properties allowing us to invoke Theorem 2.1.

**Theorem C.7.** *For every $\alpha, \varepsilon, \beta, R > 0$ and small-enough $\eta > 0$ Algorithm 2 is an $\varepsilon$-DP mechanism with the following properties. Given $n$ $\eta$-corrupted samples from any distribution $D$ with mean $\mu \in \mathbb{R}^d$ having $\|\mu\|_0 \leq k, \|\mu\| \leq R$, and covariance $\Sigma \leq I$, it produces an estimate $\hat{\mu}$ such that*

---

**Algorithm 2** Exponential-time algorithm for private sparse heavy-tailed mean estimation

---

**Input:** bucket size $b$, coordinate-wise variance $\sigma^2$, number of samples $n$

```
// Support estimation
```
1: $m_j \leftarrow \frac{1}{b} \sum_{j=i\cdot b}^{(i+1)b} x_i$ for all $j \in [\lceil n/b \rceil]$          ▷ *compute bucketed means*
2: $s_T \leftarrow S(\{m_i\}_{i \leq \lceil n/b \rceil}, T, L)$ for all $T \subseteq [d]$ with $|T| = k$          ▷ *compute scores*
3: Run the exponential mechanism [MT07] on $\{s_T\}_{T:T\subseteq[d],|T|=k}$ to get candidate support $T$
```
   // Coarse dense mean estimation (in ambient dimension k)
```
4: $S_{coarse}(\{x_{iT}\}_{i \leq n}, x, L) = \sum_{i \leq n} \mathbb{1}\{\|x_{iT} - x\| \geq L\}$
5: $s_x \leftarrow S_{coarse}(\{x_{iT}\}_{i \leq n}, x, L)$ for all $x$ in a $\sqrt{k}$-packing of the $\ell_2$ ball of radius $R$
6: Run exponential mechanism on $\{s_x\}$ to get $\hat{\mu}_{coarse}$
```
   // Fine dense mean estimation (in ambient dimension k)
```
7: $S_{fine}(\{x_{iT}\}_{i \leq n}, x, L; \hat{\mu}_{coarse}) = \max_{v:\|v\|=1} \sum_{i \leq n} \mathbb{1}\{\langle x_{iT} - x, v \rangle \geq L\}$
8: $s_x \leftarrow S_{fine}(\{m_{iT}\}_{i \leq \lceil n/b \rceil}, x, L)$ for all $x$ in a 1-packing of the $\ell_2$ ball of radius $\sqrt{k}$ centered at $\hat{\mu}_{coarse}$
9: Run exponential mechanism on $\{s_x\}$ to get $\hat{\mu}$
10: $\hat{\mu}_i \leftarrow \begin{cases} (\hat{\mu}_T)_i & \text{if } i \in T \\ 0 & \text{o.w.} \end{cases}$

**Output:** mean estimate $\hat{\mu}$

---

$\|\mu - \hat{\mu}\| \leq \alpha + O(\sqrt{\eta})$, *with probability* $1 - \beta$, *as long as*

$$n = \Omega\left(\underbrace{\frac{k \log(d) + \log(1/\beta)}{\alpha^2 \varepsilon}}_{\text{support estimation}} + \underbrace{\frac{k \log(k) + \log(1/\beta)}{\alpha^2 \varepsilon} + \frac{k \log(R)}{\varepsilon}}_{\text{dense estimation in } k \text{ dimensions}}\right).$$

To prove Theorem C.7 we can no longer rely on the hypothesis selection mechanism of [BKSW19], as the hypothesis class of distributions with bounded second moments doesn't have a finite cover in total variation distance. Instead, we design a new exponential-mechanism-based approach to identify the nonzero coordinates of $\mu$, inspired by quantile-based score functions used in robust statistics.

We show that the exponential mechanism is going to select a subset $\hat{T}$ of coordinates such that $\|\mu_{\hat{T}}\|$ is close to $\|\mu\|$ with high probability. We are guaranteed that the exponential mechanism will select a candidate with score

$$\text{OPT} - 2\frac{\Delta^{(S)}}{\varepsilon} \log\left(|\mathcal{H}|/\beta\right) \tag{7}$$

with probability at least $\beta$. We first show that the sensitivity of the score function $S$ is 1, which directly affects the guarantee in Eq. (7).

For convenience, we restate the definition of $S$ below:

$$S(\{x_i\}_{i \leq n}, T; L) = \max_{v \in \mathbb{R}^k, \|v\|_2 = 1} \sum_{i=1}^{n} \mathbb{1}\{v^\top x_{iT} \geq L\}. \tag{8}$$

To ease notation, we will omit the $\{x_i\}_{i \leq n}$ argument whenever it is clear from context.

**Lemma C.8.** *The score function $S(\cdot, \cdot; L)$ has sensitivity $\Delta^{(S)} = 1$ for all values of $L$.*

*Proof.* Changing a single element $x_i$ in the dataset affects at most one of the indicators in the sum, thus changing $S$ by at most 1. □

Next, we show that the ground-truth subset has a high score.

**Lemma C.9.** *Let $x_1, \ldots, x_n \sim P$ be independent samples and suppose that $P$ has a $k$-sparse mean $\mu$ such that $\|\mu\|_2 \leq R$, and $\|\Sigma\|_{op} \leq \lambda$. Then for the ground-truth subset $T^*$ and some $c > 0$ we have $S(\{x_i\}_{i \leq n}, T^*; \|\mu\| - c) \geq n - \frac{n\lambda}{c^2} - \sqrt{n \log(3/\beta)/2}$ with probability at least $1 - \beta/3$.*

*Proof.* Since $T^*$ is the optimal subset, we have that $\|\mu_{T^*}\| = \|\mu\|$. Using $v = \frac{\mu_{T^*}}{\|\mu_{T^*}\|}$ as a representative, we get

$$
\begin{aligned}
\mathbb{E}S(T^*; L) &\geq \mathbb{E} \sum_{i=1}^n \mathbb{1}\{\mu_{T^*}^\top x_{iT^*} \geq \|\mu_{T^*}\|L\} \\
&= \sum_{i=1}^n \mathbb{P}\left(\mu_{T^*}^\top x_{iT^*} \geq \|\mu\|L\right).
\end{aligned}
\tag{9}
$$

Now we bound $\mathrm{Var}(\mu_{T^*}^\top x_{iT^*})$. Let $\Sigma_{T^*}$ be the covariance matrix restricted to the coordinates in $T^*$. By Cauchy's interlacing theorem, we have that $\|\Sigma_{T^*}\|_{op} \leq \lambda$. Thus,

$$
\mathrm{Var}(\mu_{T^*}^\top x_{iT^*}) \leq \|\mu_{T^*}\|^2 \lambda = \|\mu\|^2 \lambda.
\tag{10}
$$

This allows us to use Chebyshev's inequality on each term of the sum to get

$$
\begin{aligned}
\mathbb{P}\left(\mu_{T^*}^\top x_{iT^*} \geq \|\mu\|L\right) &= 1 - \mathbb{P}\left(\mu_{T^*}^\top x_{iT^*} - \|\mu\|^2 < \|\mu\|L - \|\mu\|^2\right) \\
&\geq 1 - \frac{\|\mu\|^2 \lambda}{(\|\mu\|L - \|\mu\|^2)^2} \\
&= 1 - \frac{\lambda}{(L - \|\mu\|)^2} = 1 - \frac{\lambda}{c^2}
\end{aligned}
\tag{11}
$$

as long as $L < \|\mu\|$, which is guaranteed from the condition that $c > 0$. Thus, in total, we have

$$
\mathbb{E}S(T^*; \|\mu\| - c) \geq n - \frac{n\lambda}{c^2}.
\tag{12}
$$

Finally, from McDiarmid's inequality, we have that

$$
\mathbb{P}\left(S(T; L) - \mathbb{E}S(T; L) \geq t\right) \leq \exp\left(-\frac{2t^2}{n}\right),
\tag{13}
$$

which concludes the proof. $\qquad\square$

In addition to the ground-truth subset of coordinates having a large score, we want "bad" subsets to have a low score.

**Lemma C.10.** *Let $x_1, \ldots, x_n \sim P$ be independent samples and suppose that $P$ has a $k$-sparse mean $\mu$ such that $\|\mu\|_2 \leq R$, and $\|\Sigma\|_{op} \leq \lambda$. Let $T \subseteq [d]$. Then for any $L$ such that $\|\mu_T\| < L$ we have that $S(T; L) \leq n \frac{\sqrt{\lambda}}{L - \|\mu_T\|} + \frac{2}{L}\sqrt{nk\lambda} + \sqrt{n \log(3/\beta)}$ with probability at least $1 - \beta/3$.*

*Proof.* First, note that $S$ has bounded differences with respect to the sequence of random vectors $X = \{X_1, \ldots, X_n\}$. In particular, from Lemma C.8 we have that $|S(X; L) - S(X'; L)| \leq 1$ for any $X, X'$ that differ in one entry. Thus, we can apply McDiarmid's inequality to obtain

$$
\mathbb{P}\left(S(T; L) - \mathbb{E}S(T; L) \geq t\right) \leq \exp\left(-\frac{2t^2}{n}\right)
\tag{14}
$$

for all $L, T$, and $t$. The rest of the proof is devoted to bounding $\mathbb{E}S(T; L)$. To ease notation, for $X = \{x_1, \ldots, x_n\}$ define

$$
f_{T,L,v}(X) = \sum_{i=1}^n \mathbb{1}\{v^\top x_{iT} \leq L\}.
\tag{15}
$$

We start with

$$
\begin{aligned}
\mathbb{E}S(T; L) &= \mathbb{E}_X \max_{v:\, \|v\|=1} f_{T,L,v}(X) \\
&\leq \mathbb{E}_X \left[ \max_{v:\, \|v\|=1} (f_{T,L,v}(X) - \mathbb{E}f_{T,L,v}(X)) + \max_{v:\, \|v\|=1} \mathbb{E}f_{T,L,v}(X) \right] \\
&= \mathbb{E}_X \max_{v:\, \|v\|=1} (f_{T,L,v}(X) - \mathbb{E}f_{T,L,v}(X)) + \max_{v:\, \|v\|=1} \mathbb{E}f_{T,L,v}(X) \\
&= \mathbb{E}_X \max_{v:\, \|v\|=1} (f_{T,L,v}(X) - \mathbb{E}f_{T,L,v}(X)) + n \max_{v:\, \|v\|=1} \mathbb{P}_X\left(v^\top x_T \leq L\right).
\end{aligned}
\tag{16}
$$

We follow with a symmetrization argument for the first term. Let $\{\sigma_i\}_{i=1}^n$ be independent Rademacher random variables. Then

$$\mathbb{E}_X \max_{v:\, \|v\|=1} (f_{T,L,v}(X) - \mathbb{E}f_{T,L,v}(X))$$

$$\leq \mathbb{E}_{X,X',\sigma_i} \max_{v:\, \|v\|=1} (\sigma_i(f_{T,L,v}(X) - f_{T,L,v}(X')))$$

$$\leq \mathbb{E}_{X,X',\sigma_i} \max_{v:\, \|v\|=1} \left( \sum_{i\leq n} \sigma_i \left( \mathbb{1}\{v^\top x_{iT} \leq L\} - \mathbb{1}\{v^\top x'_{iT} \leq L\} \right) \right) \tag{17}$$

$$\leq 2\mathbb{E}_{X,\sigma_i} \max_{v:\, \|v\|=1} \left( \sum_{i\leq n} \sigma_i \mathbb{1}\{v^\top x_{iT} \leq L\} \right),$$

via the triangle inequality. Now we can use the inequality $\mathbb{1}\{\eta^\top v \leq L\} \leq \frac{|\eta^\top v|}{L}$ and Ledoux-Talagrand contraction [LM19] to get

$$2\mathbb{E}_{X,\sigma_i} \max_{v:\, \|v\|=1} \left( \sum_{i\leq n} \sigma_i \mathbb{1}\{v^\top x_{iT} \leq L\} \right)$$

$$\leq 2\mathbb{E}_{X,\sigma_i} \max_{v:\, \|v\|=1} \sum_{i\leq n} \sigma_i \frac{|v^\top x_{iT}|}{L}$$

$$\leq \frac{2}{L}\mathbb{E}_{X,\sigma_i} \max_{v:\, \|v\|=1} \sum_{i\leq n} \sigma_i \cdot v^\top x_{iT}$$

$$= \frac{2}{L}\mathbb{E}_{X,\sigma_i} \max_{v:\, \|v\|=1} v^\top \left( \sum_{i\leq n} \sigma_i \cdot x_{iT} \right) \tag{18}$$

$$= \frac{2}{L}\mathbb{E}_{X,\sigma_i} \left\| \sum_{i\leq n} \sigma_i \cdot x_{iT} \right\|$$

$$\leq \frac{2}{L}\sqrt{n\mathrm{Tr}(\Sigma_T)},$$

where $\Sigma_T$ is the restriction of the covariance matrix $\Sigma$ to the coordinates of the subset $T \subseteq [d]$. By the Cauchy interlacing theorem we have that $\|\Sigma_T\|_{op} \leq \|\Sigma_T\|$ and thus overall we have

$$\mathbb{E}_X \max_{v:\, \|v\|=1} (f_{T,L,v}(X) - \mathbb{E}f_{T,L,v}(X)) \leq \frac{2}{L}\sqrt{nk\lambda}. \tag{19}$$

For the second term we have

$$n \max_{v:\, \|v\|=1} \mathbb{P}_X(v^\top x_T \leq L)$$

$$= n \max_{v:\, \|v\|=1} \mathbb{P}_X(v^\top x_T - \|\mu_T\| \leq L - \|\mu_T\|)$$

$$\leq n \max_{v:\, \|v\|=1} \mathbb{P}_X(v^\top x_T - v^\top \mu_T \leq L - \|\mu_T\|) \tag{20}$$

$$= n \max_{v:\, \|v\|=1} \mathbb{E}_X \left[ \mathbb{1}\left\{ (x_T - \mu_T)^\top v \leq L - \|\mu_T\| \right\} \right]$$

$$\leq n \max_{v:\, \|v\|=1} \mathbb{E}_X \left[ \frac{|(x_T - \mu_T)^\top v|}{L - \|\mu_T\|} \right] \leq \frac{n\|\Sigma\|^{1/2}}{L - \|\mu_T\|}.$$

$\square$

Finally, we show that the dense (fine and coarse) private mean estimation algorithms satisfy the properties desired by Algorithm 2 in Lemmas C.11 and C.12.

**Lemma C.11** (Coarse dense estimation). *Let $x_1, \ldots, x_n \sim P$ be independent samples and suppose $P$ has covariance $\Sigma \leq I$ and mean $\mu \in \mathbb{R}^k$ such that $\|\mu\|_2 \leq R$. Then running exponential*

mechanism on $\{x_i\}_{i \leq n}$ with the score function $S_{coarse}(\{x_i\}_{i \leq n}, x, L) = \sum_{i \leq n} \mathbb{1}\{\|x_i - x\| \geq L\}$ gives us an estimate $\hat{\mu}$ such that $\|\hat{\mu} - \mu\|_2 \leq \sqrt{k}$ with probability at least $1 - \beta$ so long as $n = \Omega\left(\frac{k \log R + \log(1/\beta)}{\varepsilon}\right)$.

**Lemma C.12** (Fine dense estimation). *Let $x_1, \ldots, x_n \sim P$ be independent samples and suppose $P$ has covariance $\Sigma \preceq I$ and mean $\mu \in \mathbb{R}^k$ such that $\|\mu\|_2 \leq 1$ and $\|\mu\|_0 \leq k$. Then running exponential mechanism on $\{x_i\}_{i \leq n}$ with the score function $S_{fine}(\{x_i\}_{i \leq n}, x, L) = \max_{v:\|v\|=1} \sum_{i \leq n} \mathbb{1}\{\langle x_i - x, v \rangle \geq L\}$ gives us an estimate $\hat{\mu}$ such that $\|\hat{\mu} - \mu\|_2 \leq \alpha$ with probability at least $1 - \beta$ so long as $n = \Omega\left(\frac{k \log k + \log(1/\beta)}{\alpha^2 \varepsilon}\right)$.*

For both the coarse and fine estimation, our proof strategy will be as follows: first, argue that the sizes $|P_{coarse}|$ and $|P_{fine}|$ of the packings are small enough such that $\max\{\log(|P_{coarse}|), \log(|P_{fine}|)\} \leq k \log k / \alpha^2 + k \log(R)$. After that, we will show that low score (few "outliers") is achieved if and only if we are at a good mean candidate ($\alpha$-close to $\mu$).

*Proof of Lemma C.11.* The size of $\sqrt{k}$-packing of the $\ell_2$ ball of radius $R$ in $k$ dimensions is $O\left(R^k\right)$. Thus, $\log(|P_{coarse}|) \leq k \log R$, as desired.

The coarse estimation score function $S_{coarse}$ has sensitivity 1, by an argument analogous to the one in Lemma C.8.

Now we turn to showing that $S_{coarse}$ is low if and only if we are at a mean candidate $\hat{\mu}$ such that $\|\hat{\mu} - \mu\| \leq \sqrt{k}$ We can directly leverage standard concentration arguments to argue that any ball of poly$(k)$ radius that contains at least $0.9n$ of the samples must be at a distance at most $O\left(\sqrt{k}\right)$ to $\mu$ [HLZ20] with probability at least $1 - \beta$. This is enough to show both directions of the desired implication.

Thus, in total, the coarse exponential mechanism gives us a coarse estimate with good utility with probability at least $1 - \beta$ as long as $n = \Omega\left(\frac{\Delta}{\varepsilon}(k \log R + \log(1/\beta))\right) = \Omega\left(\frac{k \log R + \log(1/\beta)}{\varepsilon}\right)$. $\square$

*Proof of Lemma C.12.* As in the proof of Lemma C.11, the size of $\varepsilon$-packing (with $\ell_2$ balls) of the unit $\ell_2$ ball in $k$ dimensions is $O\left(\left(\frac{1}{\varepsilon}\right)^k\right)$. Thus, $\log(|P_{fine}|) \leq k \log k / \alpha^2$, as desired.

For the fine estimation part, without loss of generality assume that $\alpha = 1$; bucketed means as employed in the support estimation part get us at the cost of an $\alpha^2$ factor in the sample complexity. We now show that a low score for a point $x$ implies that $\|\mu - x\| \leq 1$. More formally, let "low" mean $\leq 0.1n$. A low score implies that

$$\max_{v:\|v\|=1} \langle X_i - x, v \rangle = \max_{v:\|v\|=1} (\langle X_i - \mu, v \rangle + \langle \mu - x, v \rangle) \leq L \tag{21}$$

for at least $0.9n$ indices $i$. Taking the maximum for each term, this becomes $\max_{v:\|v\|=1}\langle X_i - x, v \rangle \leq \max_{v:\|v\|=1}\langle X_i - \mu, v \rangle + \|\mu - x\|$. From the bounded covariance assumption, we get that for at least $0.9n$ of the samples, we must have $\|\mu - x\| \leq L - 1$. Setting $L = 2$ shows this direction. Suppose now we have a good mean candidate $x$. Then

$$\max_{v:\|v\|=1} \langle X_i - x, v \rangle = \max_{v:\|v\|=1} (\langle X_i - \mu, v \rangle + \langle \mu - x, v \rangle) \leq \max_{v:\|v\|=1} \langle X_i - \mu, v \rangle + 1. \tag{22}$$

Now from standard concentration we have that $\max_{v:\|v\|=1}\langle X_i - \mu, v \rangle \leq 1$ for at least $0.9n$ of the samples, and we are done.

$\square$

With this, we are ready to prove Theorem C.7.

*Proof of Theorem C.7.* First, we note that bucketing with bucket size $b$ is done in order to reduce the variance of the samples. In particular, the bucketed means $m_i$ have covariance matrix $\Sigma_m$ s.t. $\Sigma_m \preceq \lambda/bI$, where $\lambda := \lambda_{\max}(\Sigma)$.

From Eq. (7), together with Lemma C.9 and Lemma C.10, we get that to avoid choosing a subset $T$ for which $\|\mu_T - \mu\| \geq \alpha$ with probability at most $\beta$, we need

$$n - \frac{n\lambda/b}{\alpha^2} - \sqrt{\frac{n\log(3/\beta)}{2}} - \frac{2(k\log(d) + \log(3/\beta))}{\varepsilon} \geq n\frac{\sqrt{\lambda}}{2\sqrt{b}\alpha} + \frac{2}{L}\sqrt{\frac{nk\lambda}{b}} + \sqrt{\frac{n\log(3/\beta)}{2}}, \quad (23)$$

or equivalently

$$\sqrt{n}\left(1 - \frac{\lambda}{b\alpha^2} - \frac{1}{2}\sqrt{\frac{\lambda}{b\alpha^2}}\right) \geq \frac{2(k\log(d) + \log(3/\beta))}{\sqrt{n}\varepsilon} + \frac{2}{L}\sqrt{\frac{k\lambda}{b}} + 2\sqrt{\frac{\log(3/\beta)}{2}}. \quad (24)$$

This holds true when $b \geq 25\frac{\lambda}{\alpha^2}$ and
$n \geq \max\left\{10(k\log(d) + \log(3/\beta))/\varepsilon, 10k\lambda/\left(bL^2\right), 10\log(3/\beta)\right\}$. This gets us the support estimation part of the sample complexity in the statement of Theorem C.7.

Combining the support estimation step with the dense estimation steps from Lemma C.11 and Lemma C.12 gives the desired result in the absence of corruptions. The $\alpha + \sqrt{\eta}$ rate for $\eta$-corrupted inputs follows directly from Theorem 2.1. □

It is worth noting that if we directly leverage existing heavy-tailed estimators for the dense estimation subroutine [KSU20, HKM21], we would get suboptimal robustness guarantees – either $\alpha + \sqrt{d\eta}$ or $\alpha + \sqrt{\log(R)\eta}$ in the $\eta$-contamination model.

Now we turn to an information-theoretic lower bound for heavy-tailed sparse private mean estimation.

**Proposition C.13.** *Suppose $M$ is an $\varepsilon$-DP algorithm such that for every distribution $D$ with $k$-sparse mean $\mu$ with $\|\mu\| \leq R$ and covariance $\Sigma \leq I$, $M$ produces an estimate $\hat{\mu}$ from $n$ samples such that $\|\mu - \hat{\mu}\| \leq \alpha$, with probability $1 - \beta$. Then*

$$n = \Omega\left(\frac{k\log d}{\alpha^2\varepsilon}\right).$$

*Proof.* Our proof is a straightforward adaptation of the methods used to prove [KSU20, Theorem 6.1]. In particular, we show the statement is true for a class of product distributions that satisfy the mean and covariance requirements.

Formally, let

$$Q_0 = \begin{cases} -\frac{\sqrt{k}}{\alpha} & w.p. \ \frac{\alpha^2}{k} \\ 0 & w.p. \ 1 - 2\frac{\alpha^2}{k} \\ \frac{\sqrt{k}}{\alpha} & w.p. \ \frac{\alpha^2}{k}. \end{cases} \quad (25)$$

and

$$Q_1 = \begin{cases} -\frac{\sqrt{k}}{\alpha} & w.p. \ \frac{1}{2}\frac{\alpha^2}{k} \\ 0 & w.p. \ 1 - 2\frac{\alpha^2}{k} \\ \frac{\sqrt{k}}{\alpha} & w.p. \ \frac{3}{2}\frac{\alpha^2}{k}. \end{cases} \quad (26)$$

For $c \in \{0, 1\}^d$, define $Q_c = \bigotimes_{i=1}^d Q_{c_i}$. We will work only with $k$-sparse distributions, so we only need to consider $c$ such that $\|c\|_1 = k$ Let $C_k$ denote the set of such $k$-sparse vectors in $\{0, 1\}^d$. For any $c, c' \in C_k$ we have that the maximum distance in total variation between $Q_c$ and $Q_{c'}$ is at most $\alpha^2$. Now let $\mathcal{H}$ be a linear code with Hamming distance $k/4$. This implies that for every $c, c' \in \mathcal{H}$, we have that $c$ and $c'$ differ on at least $k/4$ coordinates. From coding theory (Gilbert-Varshamov bound, [BIPW10, Lemma 3.1] for a more direct argument for $k$-sparse codes) we know that there exists a code of size $|\mathcal{H}| = \Omega\left(d^k\right)$. The result follows from standard concentration and packing arguments. □

# D  Polynomial-Time Sparse Mean Estimation with $\tilde{O}(k^2)$ Samples

In this section we describe a polynomial-time $\varepsilon$-DP algorithm for sparse mean estimation with strong group privacy guarantees – it maintains privacy of groups up to size $n/\text{poly}\log(d)$. (Hence, the algorithm is automatically robust to corruption of a $1/\text{poly}\log(d)$ fraction of inputs, per Theorem 2.1.)

---

**Algorithm 3** Polynomial-time algorithm for sparse sub-Gaussian mean estimation

---

**Input:** iterations $N$, bucket size $b$, number of samples $n$, initial estimate $\mu_0$, step size $\eta$

1: $m_j \leftarrow \frac{1}{b} \sum_{j=i \cdot b}^{(i+1)b} x_i$ for all $j \in [\lceil n/b \rceil]$          ▷ *compute bucketed means*
2: **for** $1 \leq i \leq N$ **do**
3:      Check $\texttt{Halt-Estimation}\left(\{m_j\}_{j=1}^n, \mu_{i-1}\right)$, return $\mu_{i-1}$ if $\texttt{halt}$
4:      $d_i \leftarrow \texttt{Distance-Estimation}\left(\{m_j\}_{j=1}^n, \mu_{i-1}\right)$
5:      $g_i \leftarrow \texttt{Gradient-Estimation}\left(\{m_j\}_{j=1}^n, \mu_{i-1}, d_i\right)$
6:      $\bar{\mu}_i \leftarrow \mu_{i-1} + \eta d_i g_i$
7:      $\mu_i \leftarrow \texttt{Sparsify}(\bar{\mu}_i, k)$

**Output:** mean estimate $\mu_T$

---

**Theorem D.1.** *There exist universal constants $C > 0, C' > 0$ such that for every $\varepsilon, R > 0$, $\beta \in (0, 1)$, and $d, k \in \mathbb{N}$ such that $k \geq d^{0.4}$, there is a polynomial-time $\varepsilon$-DP algorithm with the following guarantees. For every $k$-sparse $\mu \in \mathbb{R}^d$ with $\|\mu\| \leq R$, given independent samples $X_1, \dots, X_n \sim \mathcal{N}(\mu, I)$, with probability at least $1 - \beta$ the algorithm outputs $x \in \mathbb{R}^d$ such that $\|\mu - x\| \leq \alpha$, so long as*

$$n \geq C(\log d)^{C'} \log R \cdot \frac{k^2 + \log(1/\beta) + \log\log R}{\alpha^2 \varepsilon} .$$

To prove Theorem D.1 we assemble our main lemmas here, and prove them in subsequent sections. The first key lemma captures a private gradient-finding procedure, finding a direction correlated with $x - \mu$ for $k$-sparse $x$.

**Lemma D.2** (Finding Gradients). *For every $c > 0$ there exists $C > 0$ such that for every $\beta, R, \varepsilon > 0$ there is a polynomial-time $\varepsilon$-DP algorithm, $\texttt{Gradient-Estimation}$, with the following guarantees. For every $d \in \mathbb{N}$ and $k \in \mathbb{N}$ with $k \geq d^{0.4}$ and every $\mu \in \mathbb{R}^d$ with $\|\mu\| \leq R$, given*

- *$n$ independent samples $X_1, \dots, X_n \sim \mathcal{N}(\mu, I)$,*

- *a vector $x \in \mathbb{R}^d$ such that*

  - *$\|x\|_2 \leq R$,*
  - *$\|x\|_0 \leq k$, and*
  - *$\|x - \mu\| \geq C$*

- *a number $r \in [(1 - 2/C)\|x - \mu\|, (1 - 1/C)\|x - \mu\|]$*

*and if*

$$n \geq (\log d)^C \cdot (k^2 + \log(1/\beta)) \cdot \frac{1}{\varepsilon} ,$$

*then with probability at least $1 - \beta$, the algorithm produces a vector $v$ such that $\left\| v - \frac{\mu - x}{\|\mu - x\|} \right\| \leq c$. Furthermore, the algorithm runs in time $\text{poly}(n, d, \log(1/\varepsilon), \log R)$.*

The second lemma shows that there is a private procedure to estimate the distance from the current iterate $x$ to $\mu$, as long as $x$ is $k$-sparse.

**Lemma D.3** (Estimating distance). *There exists a polynomial-time $\varepsilon$-DP algorithm* `Distance-Estimation` *taking as input a vector $x \in \mathbb{R}^d$ with $\|x\| \leq R$ and independent samples $X_1, \ldots, X_n$ from $\mathcal{N}(\mu, I)$, where $\|\mu\| \leq R$, with the following properties. For every $c > 0$ there is $C > 0$ such that given $n \geq \max\left\{\log(d)^C(k^2 + \log(1/\beta))/\varepsilon, 100\log(R)\log(\log(R)/\beta)/\varepsilon\right\}$ samples, and if $\|x - \mu\| \geq C$ the algorithm outputs a distance estimate $\hat{r}$ such that $(1-c)\|x - \mu\| \leq \hat{r} \leq (1+c)\|x - \mu\|$ with probability at least $1 - \beta$.*

The next lemma shows that we can preserve the sparsity of our iterates by a simple thresholding procedure.

**Lemma D.4** (Sparsifying iterates). *Let* `Sparsify(x, k)` *denote the procedure of projecting $x$ onto its $k$ largest (in absolute value) coordinates, breaking ties at random. Let $x, y \in \mathbb{R}^d$ with $\|y\|_0 \leq k$, and let $x' =$* `Sparisfy(x, k)`*. Then $\|x' - y\| \leq 4\|x - y\|$.*

The last lemma shows that the algorithm can detect when the current iterate $x$ is close enough to $\mu$.

**Lemma D.5** (Deciding to halt). *There exists a universal constant $C$ such that for all $C' > C$ there exists a polynomial-time $\varepsilon$-DP algorithm* `Halt-Estimation` *with the following guarantees. For every $k$-sparse $x$ and $\mu$ such that $\|x\|, \|\mu\| \leq R$, given $x$ and $n \geq (\log d)^C(k^2 + \log(1/\beta))/\varepsilon$ samples from $\mathcal{N}(\mu, I)$,* `Halt-Estimation` *outputs "halt" or "do not halt". If $\|x - \mu\| \leq C'$, then with probability at least $1 - \beta$,* `Halt-Estimation` *outputs "halt", and if $\|x - \mu\| \geq 200C'$ then with probability at least $1 - \beta$* `Halt-Estimation` *outputs "do not halt". Furthermore,* `Halt-Estimation` *runs in time $poly(n, d, \log(1/\varepsilon), \log R)$.*

Now we can put together the pieces to prove Theorem D.1.

*Proof of Theorem D.1.* First of all, by standard arguments, using buckets of size $b = 1/\alpha^2$ yields samples from $\mathcal{N}(\mu, \alpha^2 I)$. So by rescaling, it suffices therefore to prove the theorem in the case $\alpha = \Theta(1)$, which is what we will do.

For some $\beta'$ we choose momentarily, let us condition on all of the $1 - \beta'$-probability events specified in Lemmas D.2, D.3, and D.5, for all $O(\log R)$ calls to `Halt-Estimation`, `Distance-Estimation`, and `Gradient-Estimation`. Choosing $\beta' = \Omega(\beta/\log R)$, so long as $n \geq (\log d)^{O(1)} \log R(k^2 + \log(1/\beta) + \log\log R)/\varepsilon$, by a union bound, with probability $1 - \beta$ all these events occur.

Having conditioned on all these events, from Lemma D.5, it suffices to show that there exists $C > 0$ such that within $O(\log R)$ iterations of the main loop of Algorithm 3 some iterate $\mu_i$ has $\|\mu_i - \mu\| \leq C$. Then, we can choose $C'$ in Lemma D.5 to have $C' \geq C$ and we obtain the guarantee that (a) `Halt-Estimation` outputs "halt" in some iteration and (b) when it does so, the output $\mu_i$ satisfies $\|\mu_i - \mu\| \leq O(1)$.

It suffices therefore to show that there exists universal constant $C' > 0$ such that each iteration of gradient descent (lines 4-7 in Algorithm 3) shrinks $\|\mu_t - \mu\|$ by a factor of at least 0.9, i.e. if $\|\mu_{t-1} - \mu\| \geq C'$ then $\|\mu_t - \mu\| \leq 0.9\|\mu_{t-1} - \mu\|$. Now we show that this follows from the guarantees of Lemma D.2 and Lemma D.3. In particular, it is enough to show that $\left\|g_t - \frac{\mu - \mu_{t-1}}{\|\mu - \mu_{t-1}\|}\right\| \leq 0.01$. Given that $(1-c)\|\mu_{t-1} - \mu\| \leq d_t \leq (1+c)\|\mu_{t-1} - \mu\|$ for a constant $c$ we will choose below (Lemma D.3), before calling `Sparsify` we have that

$$
\begin{aligned}
\|\bar{\mu}_t - \mu\| &= \left\|\left(\mu_{t-1} - \mu + \eta\frac{\mu - \mu_{t-1}}{\|\mu - \mu_{t-1}\|}d_t\right) + \eta d_t g_t - \eta d_t\frac{\mu - \mu_{t-1}}{\|\mu - \mu_{t-1}\|}\right\| \\
&\leq \left\|(\mu_{t-1} - \mu) - \eta\frac{\mu_{t-1} - \mu}{\|\mu - \mu_{t-1}\|}d_t\right\| + 0.01\eta d_t \\
&\leq \left\|1 - \frac{\eta d_t}{\|\mu_{t-1} - \mu\|}\right\|\|\mu_{t-1} - \mu\| + 0.01(1+c)\eta\|\mu_{t-1} - \mu\| \\
&\leq \left(\max\{|1 - \eta(1+c)|, |1 - \eta(1-c)|\} + 0.01(1+c)\eta\right)\|\mu_{t-1} - \mu\|.
\end{aligned}
\tag{27}
$$

Setting $\eta = 1$, we get that

$$
\|\bar{\mu}_t - \mu\| \leq (1.01c + 0.01)\|\mu_{t-1} - \mu\|.
\tag{28}
$$

Choosing $c \leq 0.1$, and applying the guarantees of Lemma D.4 (`Sparsify` loses at most a factor of 4) finishes the argument. $\qquad\square$

# E   Omitted Proofs from Appendix D

To set up, we define two key systems of polynomial inequalities.

**Definition E.1** (The polynomial systems $\mathcal{Q}$ and $\mathcal{P}$). *Let $\mathcal{Q}$ be the following polynomial system in variables $v_1, \ldots, v_d, c_1, \ldots, c_d$:*

$$\|v\|^2 = 1, \ \|c\|^2 = 1, \ v_i v_j \leq c_i c_j \text{ and } -v_i v_j \leq c_i c_j \text{ for all } i, j \in [d], \ \left( \sum_{i \in [d]} c_i \right)^2 \leq s.$$

*For $X_1, \ldots, X_n \in \mathbb{R}^d$ and $x \in \mathbb{R}^d$ and $\alpha, s > 0$, we define the following system of inequalities $\mathcal{P}(x, \delta, \alpha, s, X_1, \ldots, X_n)$ in variables $w_1, \ldots, w_d, v_1, \ldots, v_d, b_1, \ldots, b_n, c_1, \ldots, c_d$ to be:*

$$\mathcal{Q} \cup \{b_i^2 = b_i, b_i \langle X_i - x, v \rangle \geq b_i \alpha\}_{i \in [n]} \cup \{\|w - v\|^2 \leq \delta\}.$$

## E.1   Estimating Distance

Here we prove Lemma D.3.

*Proof.* Let $X = \{X_i\}_{i \leq n}$ and let $f(r, X; \delta, s) = \max \tilde{\mathbb{E}} \sum_{i \leq n} b_i$ s.t. $\tilde{\mathbb{E}}$ satisfies $\mathcal{P}(x, \delta, r, s, X)$; in words, $f(r, X)$ (roughly) counts the number of samples considered as "inliers" when we use radius $r$. The key idea is that this number is "large" when $r \gg \|x - \mu\|$ and "small" when $r \ll \|x - \mu\|$.

More formally, we invoke the following private binary search primitive [HKM21, Theorem 6.15]: Given a function $f : [0, R] \times \mathcal{X}^n \to [0, n]$, which is decreasing in its first argument, and has sensitivity 1 in its second argument, $S \stackrel{\text{def}}{=} \log(R/a)$ rounds of binary search suffice to output an estimate $\hat{r} \in [r_e - a, r_s + a]$, where $r_e = \min r$ s.t. $f(r, X) \leq ek + \Delta$, and $r_s = \max r$ s.t. $f(r, X) \geq sk - \Delta$ for $\Delta = S \log(S/\beta)/\varepsilon$. We choose $s = 1 - 2\delta, e = 1 - \delta, \Delta = \frac{\delta}{10} n$ for some $\delta > 0$.

First, monotonicity follows from the fact that the solution for a given $r = r' > r''$ is a feasible solution for $r = r''$, and bounded sensitivity follows from the fact that the value of the (non-relaxed) quadratic program has sensitivity 1 since we can change at most one of the indicators upon changing a single sample; furthermore, there is an SoS proof of that [HKM21].

Now we turn to quantifying $r_e$ and $r_s$; we show the following two facts:

1. For all $c_e > 0, \delta > 0$, there exists $C_e > 0$ such that if $\|x - \mu\| \geq C_e$, then $r_e \geq (1 - c_e)\|x - \mu\|$.

2. For all $c_s > 0, \delta > 0$, there exists $C_s > 0$ such that if $\|x - \mu\| \geq C_s$, then $r_s \leq (1 + c_s)\|x - \mu\|$.

For Fact 1, it suffices to show that $f((1 - c_e)\|x - \mu\|, X) \geq (1 - \delta)n$. Now, it is enough to exhibit a feasible direction. In particular, set $v = \frac{\mu - x}{\|\mu - x\|}$. Let $S \subseteq [n]$ be a subset of samples such that $|S| \geq (1 - \delta)n$ and $\mathcal{Q} \vdash_{O(1)} \sum_{i \in S} \langle X_i - \mu, v \rangle^2 \leq 2n$. We know such a subset exists from Lemma E.3 as long as $n \geq \log(d)^C \left( k^2 + \log(1/\beta) \right)/\varepsilon$. Then we have that there exists $S' \subseteq S$ such that $|S'| \geq (1 - 2\delta)n$ for which we have $|\langle X_i - \mu, v \rangle| \leq 1/(2\delta)$ for all $i \in S'$. From this, we know that

$$\sum_{i \in S'} \langle X_i - x, v \rangle = \sum_{i \in S'} \langle X_i - \mu, v \rangle + \sum_{i \in S'} \langle \mu - x, v \rangle \geq -\frac{1}{2\delta} + \|x - \mu\|. \tag{29}$$

Now we need $\|x - \mu\| \geq \frac{1}{2c_e \delta}$ for Fact 1 to hold. Setting $C_e$ to the RHS finishes the argument.

For Fact 2, it is enough to show that $f((1 + c_s)\|x - \mu\|, X) \leq (1 - 2\delta)n$. Fix $\delta'$ and $L$ to be chosen later. Let $S \subseteq [n]$ be a subset of samples such that $|S| \geq (1 - \delta')n$ and $\mathcal{Q} \vdash_{O(1)} \sum_{i \in S} \langle X_i - \mu, v \rangle^2 \leq 2n$ (such exists from Lemma E.3).

Note that $(a + b)^2 \leq (1 + L^2)a^2 + (1 + 1/L^2)b^2$ is true for any $L \in \mathbb{R}$ and has an SoS proof. Using this and the SoS Cauchy-Schwarz inequality, we have the following sequence of inequalities, all with

degree-2 SoS proofs.

$$\left(\sum_{i\leq n}b_i\right)^2 \leq \left((n-|S|)+\sum_{i\in S}b_i\right)^2 \leq \left((n-|S|)+\sum_{i\in S}b_i\frac{\langle X_i-x,v\rangle}{r}\right)^2$$

$$\leq (1+L^2)(n-|S|)^2 + \left(1+\frac{1}{L^2}\right)\left(\sum_{i\in S}b_i\frac{\langle X_i-x,v\rangle}{r}\right)^2$$

$$\leq (1+L^2)(\delta'n)^2 + \left(1+\frac{1}{L^2}\right)\left(\sum_{i\in S}b_i\frac{\langle X_i-\mu,v\rangle+\langle\mu-x,v\rangle}{r}\right)^2$$

$$\leq (1+L^2)(\delta'n)^2+$$

$$\left(1+\frac{1}{L^2}\right)\left((1+L^2)\left(\sum_{i\in S}b_i\frac{\langle X_i-\mu,v\rangle}{r}\right)^2 + \left(1+\frac{1}{L^2}\right)\left(\sum_{i\in S}b_i\frac{\langle\mu-x,v\rangle}{r}\right)^2\right) \quad (30)$$

$$\leq (1+L^2)(\delta'n)^2 + \left(1+\frac{1}{L^2}\right)(1+L^2)\left(\sum_{i\in S}b_i^2\right)\frac{2n}{r^2}+$$

$$\left(1+\frac{1}{L^2}\right)^2\left(\sum_{i\in S}b_i^2\right)\left(\sum_{i\in S}\frac{\|\mu-x\|^2}{r^2}\right)$$

$$\leq n^2\cdot\left((1+L^2)(\delta')^2 + \left(L^2+2+\frac{1}{L^2}\right)\frac{|S|}{n}\frac{2}{r^2} + \left(1+\frac{1}{L^2}\right)^2\frac{|S|^2}{n^2}\frac{\|\mu-x\|^2}{r^2}\right)$$

$$\leq n^2\cdot\left((1+L^2)(\delta')^2 + \left(2L^2+4+\frac{2}{L^2}+\left(1+\frac{1}{L^2}\right)^2\|\mu-x\|^2\right)\frac{1}{r^2}\right)$$

We can choose $\delta' \leq \frac{\sqrt{1-2\delta}}{100\sqrt{1+L^2}}$ such that the first term is sufficiently small. Next, we choose $L$ so that the overall sum is at most $(1-2\delta)n$. In particular, we want that

$$\frac{2L^2+4+\frac{2}{L^2}}{(1+c_s)^2\|\mu-x\|^2} + \left(\frac{1+1/L^2}{1+c_s}\right)^2 \leq \frac{2L^2+4+\frac{2}{L^2}}{(1+c_s)^2C_s^2} + \left(\frac{1+1/L^2}{1+c_s}\right)^2 \leq (1-2\delta)^2. \quad (31)$$

Thus, we can choose $L$ such that $L \geq \frac{1}{\sqrt{c_s-2\delta(1+c_s)}}$ and $2L^2+4+2/L^2 \leq ((1-2\delta)(1+c_s)C_s)^2$ (solving self-consistently).

Combining the conclusion of [HKM21, Theorem 6.15] with the conclusions of Facts 1 and 2, we receive as output a distance estimate $\hat{r} \in [(1-c_e)\|x-\mu\|-a, (1+c_s)\|x-\mu\|+a]$. Noting that $\|x-\mu\| \geq \max\{C_e, C_s\}$, we can set $c_s = c_e = c/2$ and $a = \max\{C_e, C_s\}\cdot c/2$ to obtain the desired result.

□

## E.2 Sparsifying

**Lemma E.2** (Lemma D.4 restated). *Let* $\mathtt{Sparsify}(x,k)$ *denote the procedure of projecting* $x$ *onto its* $k$ *largest (in absolute value) coordinates, breaking ties at random. Let* $x,y \in \mathbb{R}^d$ *with* $\|y\|_0 \leq k$, *and let* $x' = \mathtt{Sparisfy}(x,k)$. *Then* $\|x'-y\| \leq 4\|x-y\|$.

*Proof.* Let $S = \mathrm{supp}(y)$ and $T = \mathrm{supp}(x')$. Now note that $|S \setminus T| = |T \setminus S| \leq k$. Additionally, from the assumption that we select the largest $k$ components, we have that $\|x_{T\setminus S}\| \geq \|x_{S\setminus T}\|$.

Using the above observation, together with the triangle inequality, we have that

$$
\begin{aligned}
\|x' - y\| &\leq \|x'_{S\setminus T} - y_{S\setminus T}\| + \|x'_{T\setminus S} - y_{T\setminus S}\| + \|x - y\| \\
&\leq \|x'_{S\setminus T} - x_{S\setminus T}\| + \|x_{S\setminus T} - y_{S\setminus T}\| + \|x_{T\setminus S} - y_{T\setminus S}\| + \|x - y\| \\
&\leq \|x_{S\setminus T}\| + 3\|x - y\| \\
&\leq \|x_{T\setminus S}\| + 3\|x - y\| \\
&\leq 4\|x - y\|,
\end{aligned}
\tag{32}
$$

as desired.

$\square$

## E.3 Deciding to Halt

In this section we prove Lemma D.5.

*Proof of Lemma D.5.* The algorithm is as follows. Compute $Z \overset{\text{def}}{=} \max \tilde{\mathbb{E}} \sum_{i=1}^{n} b_i$ over all $\tilde{\mathbb{E}}$ satisfying $\mathcal{P}(x, 1, 100C', X_1, \ldots, X_n)$ and add noise $Lap(1/\varepsilon)$. If the result is $\leq 0.1n$, output "halt", otherwise output "do not halt".

Sensitivity of $Z$ was already proved in Lemma D.3. Privacy follows directly from the guarantees of the Laplace mechanism.

Next we establish correctness. This also follows directly from the arguments in Lemma D.3 since we can view halt estimation as a one-step binary search. In particular, let $S \subseteq [n]$ be a subset such that $|S| \geq (1 - \delta)n$ and $Q \vdash \sum_{i \in S}\langle X_i - x, v\rangle^2 \leq 2n$. Then we claim that the algorithm above will output "halt" with probability at least $1 - \beta$, using that $C'$ is at least some universal constant.

At the same time, suppose that the algorithm above outputs "halt". Conditioning again on an event of probability $1 - \beta$, this means that $\max \tilde{\mathbb{E}} \sum_{i=1}^{n} b_i$ over all $\tilde{\mathbb{E}}$ satisfying $\mathcal{P}(x, 1, 100C', X_1, \ldots, X_n)$ is at most $0.2n$. In particular, for every $2k$-sparse unit vector $v$, at least $0.8n$ choices of $i$ have $\langle X_i - x, v\rangle \leq 100C'$, and the same is true for $\langle X_i - \mu, v\rangle$. Choosing some $i$ such that this holds simultaneously for $v = (\mu - x)/\|\mu - x\|$ in the first case and $v = (x - \mu)/\|x - \mu\|$ in the second, we obtain by adding the two,

$$
200C' \geq \langle X_i - x, (\mu - x)/\|\mu - x\|\rangle - \langle X_i - \mu, (\mu - x)/\|\mu - x\|\rangle = \|\mu - x\|.
$$

$\square$

## E.4 Finding Gradients

In this section we prove Lemma D.2. The promised algorithm will use the SoS exponential mechanism of [HKM21].

Our first lemma says that the quadratic form of the covariance matrix of $X_1, \ldots, X_n$, after throwing out a few samples, has bounded quadratic form in sparse directions, and that furthermore this has an SoS proof.

**Lemma E.3.** *For every $c > 0$ there exists a universal constant $C > 0$ such that for all large-enough $d \in \mathbb{N}$, all $s \leq d$, and all $\beta \in (0, 1)$, for all $n \geq (\log d)^C(s^2 + \log(1/\beta))$, with probability at least $1 - \beta$ over i.i.d. $X_1, \ldots, X_n \sim \mathcal{N}(0, I)$ there is a subset $S \subseteq [n]$ with $|S| \geq (1 - c)n$ and such that*

$$
Q \vdash_{O(1)} \sum_{i \in S}\langle X_i, v\rangle^2 \leq 2n,
$$

The next lemma gives the SoS proof of utility we need for SoS exponential mechanism.

**Lemma E.4.** *For every $c > 0$ there is $C > 0$ such that the following holds. Let $X_1, \ldots, X_n \sim \mathcal{N}(\mu, I)$ for $\mu \in \mathbb{R}^d$ with $\|\mu\|_0 \leq k$, with $d \in \mathbb{N}$ sufficiently large. For all $x \in \mathbb{R}^d$ with $\|x - \mu\| > C$, all $\beta \in (0, 1)$, and $r \in [(1 - 1/C)\|x - \mu\|, (1 + 1/C)\|x - \mu\|]$, and*

$n \geq (\log d)^C (k^2 + \log(1/\beta))$, *with probability at least* $1 - \beta$ *over* $X_1, \ldots, X_n$, *there is a degree-*$O(1)$ *SoS proof*

$$\mathcal{P}(x, 1/C, r, k/c, X_1, \ldots, X_n), \sum_{i \leq n} b_i \geq (1 - 1/C)n \vdash_{O(1)} \left\| w - \frac{\mu - x}{\|\mu - x\|} \right\|^2 \leq c \, ,$$

*and, furthermore, this proof has degree-*1 *in the constraint* $\sum_{i \leq n} b_i \geq (1 - 1/C)n$.

The last lemma proves key properties of the convex set over which SoS exponential mechanism will run a log-concave sampling algorithm.

**Lemma E.5.** *Let* $d, k \in \mathbb{N}$ *with* $k \leq d$, *and let* $\delta > 0$. *Then the set*

$$C(k, \delta) = \{x \in \mathbb{R}^d \ : \ \exists y \in \mathbb{R}^d \text{ s.t. } \|y\|_2 \leq 1, \|y\|_1 \leq \sqrt{k}, \|x - y\|_2 \leq \delta\}$$

*has the following properties:*

- $C$ *is compact and convex.*

- $C$ *has diameter* $O(1 + \delta)$.

- $C$ *admits polynomial-time projection and membership oracles.*

- *For every* $v \in \mathbb{R}^d$ *with* $\|v\| \leq 1$ *and* $\|v\|_1 \leq \sqrt{k}$, *there exists a set* $S_v \subseteq C$ *such that for all* $w \in S_v$, $\|w - v\| \leq \delta$, *and* $|S_v| \geq \exp(-O(d/(\delta \sqrt{k}) + k^2 \log d)) \cdot |C|$, *where* $|\cdot|$ *denotes Lebesgue measure.*

Now we can apply Theorem 4.5 of [HKM21] to prove Lemma D.2.

*Proof of Lemma D.2.* We verify that the conditions of Theorem 4.5 of [HKM21] apply to our polynomial system $\mathcal{P}$, with $p(b) = \sum_{i \in [n]} b_i - (1 - 1/C)n$ for a large-enough constant $C$, and the convex set $C(k, 1/C)$ of Lemma E.5.

- Compactness and convexity of $C$, projection and membership oracle, diameter at most $\text{poly}(n, d)$: guaranteed by Lemma E.5.

- $\mathcal{P}$ is Archimedian: $\mathcal{P}$ contains constraints upper-bounding $\|v\|^2, \|c\|^2, \|b\|^2$ by $\text{poly}(d, n)$, by inspection.

- Robust satisfiability: let $\eta = 1/\text{poly}(n, d)$. We claim that for all $X_1, \ldots, X_n, \alpha$, the system $\mathcal{P}(x, 2/C, \alpha, s, X_1, \ldots, X_n)$ is $\eta$-robustly satisfiable with respect to $C$ and $p(b) = \sum_{i \leq n} b_i - (1 - 1/C)n$.

  To see this, consider any $y \in C$ and any $y'$ such that $\|y' - y\| \leq 1/\text{poly}(d, n)$. Then, setting $w = y'$, we can satisfy $\mathcal{P}$ by choosing $v$ to be the element of $C$ satisfying $\|v - y\| \leq 1/C$ and having $\|v\|_2 \leq 1, \|v\|_1 \leq \sqrt{k/c}$; then we will have $\|w - v\| \leq 1/C + 1/\text{poly}(d, n) \leq 2/C$. Choosing $c_i = |v_i|$, and choosing $b_i = 0$ for all $i$ satisfies the rest of the constraints in $\mathcal{P}$.

- SoS proof of bounded sensitivity: same as Example 4.2 in [HKM21].

- SoS proof of utility: existence with probability at least $1 - \beta$ over choice of samples follows directly from Lemma D.2. This proof if expressible in at most $\text{poly}(d, n, \log R)$ bits by inspection of the proof of Lemma D.2.

- Volume ratio: We claim that for $n \geq (\log d)^C (k^2 + \log(1/\beta))$ and $k \geq d^{0.4}$, there exists a set $S \subseteq C$ with $|S|/|C| \geq d^{-O(k^2)}$ (where $|\cdot|$ denotes Lebesgue measure) such that for all $z \in S$ there is a solution $v, w, b, c$ to $\mathcal{P}$ with $p(b) \geq \Omega(n)$ and $w = z$.

  To see this, we take $S$ to be the set of $z$ such that $\|z - (\mu - x)/\|\mu - x\|\| \leq 1/C$. First, $\|\mu - x\|_0 \leq 2k$, so $\|\mu - x\|_1 \leq \sqrt{2k}\|\mu - x\|_2$, hence $S \subseteq C$. For any $z \in S$ we can set $w = z$ and $v = (\mu - x)/\|\mu - x\|$, then $\|w - v\|^2 \leq 2/C$. Take $c_i = |v_i|$; since $v$ is

$2k$-sparse this satisfies the constraints of $Q$. Take $b_i = 1$ if
$\langle X_i - x, \frac{\mu - x}{\|\mu - x\|} \rangle = \langle X_i - \mu, \frac{\mu - x}{\|\mu - x\|} \rangle + \|\mu - x\| \geq r$, again satisfying $\mathcal{P}$.

It remains to show that $\sum_{i \in [n]} b_i - (1 - 1/C)n \geq \Omega(n)$. Applying Lemma E.3 and Markov's inequality, for large-enough $n \gg (\log d)^{C'}(k^2 + \log(1/\beta))$ for some other constant $C'$, there is a set $T \subseteq [n]$ with $|T_i| \geq (1 - 1/(10C))n$ and $|\langle X_i - \mu, (\mu - x)/\|\mu - x\|\rangle| \leq O(1)$. So, as long as $r \leq \|\mu - x\| - O(1)$, $\sum b_i$ is large enough.

It follows that for fixed $x$ and $\mu$, with probability at least $1 - \beta$ over $n \geq (\log d)^C(k^2 + \log(1/\beta))/\varepsilon$ samples from $\mathcal{N}(\mu, I)$, SoS exponential mechanism is $\varepsilon$-DP and outputs a vector $v$ such that $\|v - (\mu - x)/\|\mu - x\|\| \leq c$.

For the running time dependence on $\log(1/\varepsilon)$, note that the dependence of $1/\varepsilon$ claimed in [HKM21] for SoS exponential mechanism can be improved by appeal to the main result of [MV21]. $\square$

### E.4.1 Proof of Lemma E.4, Lemma E.3

*Proof of Lemma E.3.* Consider the random variable $B = \max\{|S| : Q \vdash_{O(1)} \sum_{i \in S} \langle X_i, v \rangle^2 \leq 2n\}$. By McDiarmid's bounded-differences inequality, $\Pr(|B - \mathbb{E}B| > t) \leq 2\exp(-\Omega(t^2/n))$, so as long as $n \gg \log(1/\beta)$, we have $|B - \mathbb{E}B| \leq cn/10$ with probability at least $1 - \beta$. It will suffice therefore to show that $\mathbb{E}B \geq (1 - c/10)n$. For this in turn it suffices to show that with probability at least $1 - c/20$ we have $B \geq (1 - c/20)n$.

Let $S \subseteq [n]$ be the set of $X_i$ such that $\|X_i\|_\infty \leq c'\sqrt{\log d}$, with $c' > 0$ chosen so that $|S| \geq (1 - c/20)n$ with probability at least $(1 - c/100)$. We claim that if $n \gg \text{poly}\log(d) \cdot s^2$ then $Q \vdash_{O(1)} \sum_{i \in S} \langle X_i, v \rangle^2 \leq 2n$ with probability at least $1 - c/100$; then a union bound finishes the proof.

Let $M = \sum_{i \in S} X_i X_i^\top$. By Bernstein's inequality, with probability at least $1 - \delta$ the following both hold:

$$\max_{a \neq b \in [d]} |M_{ab}| \leq O(\sqrt{n\log(1/\delta)} + \log d \cdot \log(1/\delta))$$

$$\max_{a \in [d]} |M_{aa}| \leq n + O(\sqrt{n\log(1/\delta)} + \log d \cdot \log(1/\delta)).$$

For all $a, b \in [d]$, $Q \vdash_{O(1)} v_a v_b M_{ab} \leq |M_{ab}|c_a c_b$, so

$$Q \vdash \sum_{i \leq n} \langle X_i, v \rangle^2 \leq \max_{a \in [d]} |M_{aa}| \sum_{a \in [d]} c_a^2 + \max_{a \neq b \in [d]} |M_{ab}| \cdot \left(\sum_{a \in [d]} c_a\right)^2$$

$$\leq \max_{a \in [d]} |M_{aa}| + s \cdot \max_{a \neq b \in [d]} |M_{ab}|$$

$$\leq s \cdot O(\sqrt{n\log(1/\delta)} + \log d \cdot \log(1/\delta)) + n.$$

Choosing $\delta$ a small enough, and then $C$ large enough, completes the proof. $\square$

**Lemma E.6.** *For an indeterminate $X$ and $C > 0$,*
$$X^2 \leq C^2 \vdash_2 X \leq 2C.$$

*Proof.* Note that $X = \frac{1}{2C}((X + C)^2 - X^2 - C^2)$. So $X^2 \leq C^2 \vdash X \leq \frac{1}{2C}(X + C)^2 \leq \frac{1}{2C} \cdot 2(X^2 + C^2)$ via SoS triangle inequality. Using the axiom $X^2 \leq C^2$ completes the proof. $\square$

Now we can prove Lemma E.4.

*Proof of Lemma E.4.* First, by SoS triangle inequality and the constraint $\|w - v\|^2 \leq 1/C$, for large-enough $C$, it suffices to show that

$$\mathcal{P}(x, 1/C, r, k, X_1, \ldots, X_n), \sum_{i \leq n} b_i \geq (1 - 1/C)n \vdash_{O(1)} \left\|v - \frac{\mu - x}{\|\mu - x\|}\right\|^2 \leq c/4.$$

And, for this, it is enough to show $\mathcal{P}, \sum_{i\le n} b_i \ge (1 - 1/C)n \vdash_{O(1)} \langle \mu - x, v\rangle \ge (1 - c/8)\|\mu - x\|$. (All these proofs must be degree-1 in $\sum_{i\le n} b_i \ge (1 - 1/C)n$, which will be true by construction.)

Let $S \subseteq [n]$ be the set of indices guaranteed to exist by Lemma E.3, of size $(1 - 1/C)n$, for the vectors $X_1 - \mu, \ldots, X_n - \mu$. We first claim that

$$\mathcal{P} \vdash_{O(1)} \sum_{i\in S} b_i \langle X_i - \mu, v\rangle \le 4n \,.$$

For this, using Lemma E.6, we can show instead

$$\mathcal{P} \vdash_{O(1)} \left(\sum_{i\in S} b_i \langle X_i - \mu, v\rangle\right)^2 \le 2n^2 \,,$$

but this follows immediately from SoS Cauchy-Schwarz and Lemma E.3.

Now,

$$\mathcal{P}, \sum_{i\in n} b_i \ge (1 - 1/C)n \vdash_{O(1)} r \cdot (1 - 2/C)n \le r \cdot \sum_{i\in S} b_i$$

$$\le \sum_{i\in S} b_i \langle X_i - x, v\rangle = \sum_{i\in S} b_i \langle X_i - \mu, v\rangle + b_i \langle \mu - x, v\rangle \,.$$

Putting this together with the preceding, we have

$$\mathcal{P}, \sum_{i\in[n]} b_i \ge (1 - 1/C)n \vdash_{O(1)} (1 - 2/C)rn - 4n \le \langle \mu - x, v\rangle \cdot \sum_{i\in S} b_i = n \cdot \langle \mu - x, v\rangle + \left(\sum_{i\in S} b_i - n\right) \cdot \langle \mu - x, v\rangle \,.$$

Next we claim that
$\mathcal{P}, \sum_{i\in[n]} b_i \ge (1 - 1/C)n \vdash_{O(1)} (\sum_{i\in S} b_i - n) \cdot \langle \mu - x, v\rangle \le O(1/\sqrt{C})n\|\mu - x\|$.
We have

$$\|v\|^2 \le 1 \vdash_{O(1)} \left(\sum_{i\in S} b_i - n\right) \cdot \langle \mu - x, v\rangle$$

$$\le \frac{\sqrt{C}\|\mu - x\|}{n} \cdot \left(\sum_{i\in S} b_i - n\right)^2 + \frac{n}{\sqrt{C}\|\mu - x\|}\langle \mu - x, v\rangle^2$$

$$\le \frac{\sqrt{C}\|\mu - x\|}{n} \cdot \left(\sum_{i\in S} b_i - n\right)^2 + (1/\sqrt{C})n\|\mu - x\| \,.$$

Now, $\{b_i^2 = b_i\}_{i\in[n]} \vdash_{O(1)} (\sum_{i\in S} b_i - n)^2 \le 2n (\sum_{i\in S} b_i - n)$, so via a proof which is degree-1 in $\sum_{i\in[n]} b_i \ge (1 - 1/C)n$ we have

$$\mathcal{P}, \sum_{i\in[n]} b_i \ge (1 - 1/C)n \vdash_{O(1)} \left(\sum_{i\in S} b_i - n\right)^2 \le 2 \cdot 1/C \cdot n^2 \,.$$

Putting everything together and using the assumption $r \ge (1 - 1/C)\|\mu - x\|$, we get

$$\mathcal{P}, \sum_{i\le n} b_i \ge (1 - 1/C)n \vdash_{O(1)} (1 - O(1/\sqrt{C}))\|\mu - x\| - 4 \le \langle \mu - x, v\rangle$$

which gives the conclusion by taking $C = C(c)$ large enough. $\qquad\square$

### E.4.2   Proof of Lemma E.5

*Proof of Lemma E.5.* Let $C = \{x \in \mathbb{R}^d \ : \ \exists v \in \mathbb{R}^d \text{ s.t. } \|x - v\| \le \delta \text{ and } \|v\|_1 \le \sqrt{k}, \|v\|_2 \le 1\}$ be a "fattening" of the scaled $\ell_1$ ball. $C$ is convex by inspection.

To compute membership in $C$, given $x$, compute the projection $x'$ to $\{v : \|v\|_1 \leq \sqrt{k}$ and $\|v\|_2 \leq 1\}$. If $\|x - x'\| \leq \delta$ then $x \in C$ and otherwise $x \notin C$. Furthermore, if $\|x - x'\| > \delta$, then for some $\delta'$, a hyperplane through $x + \delta'(x' - x)$ separates $x$ from $C$. With this separation oracle for $C$, projections can be computed by minimizing $\|x - y\|$ over $y \in C$.

Now we move on to the volume arguments. By Sudakov minoration (see e.g. [LT91]), $\{v \ \|v\|_1 \leq \sqrt{k}\}$ can be covered by $d^{O(k^2)}$ $\ell_2$ balls of radius $1/\sqrt{k}$. So, $C$ can be covered by $d^{O(k^2)}$ $\ell_2$ balls of radius $1/\sqrt{k} + \delta$. For any $v$ with $\|v\|_1 \leq \sqrt{k}$ and $\|v\|_2 \leq 1$, we know $S_v = \{w : \|w - v\| \leq \delta\} \subseteq C$. Furthermore,

$$\frac{|S_v|}{|C|} \geq d^{-O(k^2)} \cdot \left(\frac{\delta}{\delta + \frac{1}{\sqrt{k}}}\right)^d = d^{-O(k^2)} 2^{-O(d/(\sqrt{k}\delta))}.$$

$\square$

# F Proof of Theorem 4.1

*Proof.* The key idea in our analysis is to show that there exists a parameter regime (for the threshold $T$ and bucket size $b$) for which there is a separation between the minimum score $z_i$ of coordinates for which $|\mu_i| > T$ and the maximum expected score of coordinates for which $\mu_i = 0$.

Assume for simplicity that $n/b = \lceil n/b \rceil$. Note that $m_i \sim \mathcal{N}(\mu, \sigma^2/bI)$. Define $p_i := \mathbb{P}(|(m_1)_i| > T)$ as the probability of a (bucketed mean of a) sample exceeding the given threshold $T$. Then $z_i \sim \mathrm{Bin}(n/b, p_i)$. Note that from Chernoff we have for all coordinates $i$ outside the support of $\mu$ that $p_i \leq \exp\left(-bT^2/(2\sigma^2)\right) =: p_0$. Let $z_0 \sim \mathrm{Bin}(n/b, p_0)$. Since all $p_i$s are bounded below by a constant, we can use the normal approximation to $z_i$ and sub-Gaussian concentration. On one hand,

$$\mathbb{E} \max_{i:\mu_i=0} z_i \leq \frac{n}{b} p_0 + \sqrt{2(\log d + \log(4k/\beta)) \cdot \mathrm{Var}(z_0)}$$

$$\leq \frac{n}{b} \exp\left(\frac{-bT^2}{2\sigma^2}\right)\left(1 + \sqrt{\frac{2b(\log d + \log(4k/\beta))}{np_0}}\right) \tag{33}$$

with probability at least $1 - \beta/(4k)$.

On the other, for large coordinates $i$ where $\mu_i \geq T$, we have that $p_i \geq \frac{1}{2}$ and thus

$$\mathbb{E} \min_{i:|\mu_i| \geq T} z_i \geq \frac{n}{b} \min_{i:|\mu_i| \geq T} p_i - \sqrt{2(\log d + \log(4k/\beta)) \cdot (n/b) \cdot \min_{i:|\mu_i| \geq T} p_i(1 - p_i)}$$

$$\geq \frac{n}{b}\left(\frac{1}{2} - \sqrt{\frac{2b(\log d + \log(4k/\beta))}{n}}\right), \tag{34}$$

again with probability at least $1 - \beta/(4k)$.

We can ensure that $\mathbb{E} \min_{i:|\mu_i| \geq T} z_i - \frac{2k(\log(d) + \log(4k/\beta))}{\varepsilon} > \mathbb{E} \max_{i:\mu_i=0} z_i$, by setting $T = 3.5\sigma/\sqrt{b}$ and $n \geq \max\{1, k/\varepsilon\} \cdot 20b(\log(d) + \log(4k/\beta))$. For all practical purposes, we can safely assume $k/\varepsilon \geq 1$ and thus get $n \geq 20kb(\log(d) + \log(4k/\beta))/\varepsilon$. For all

In $k$ rounds of the exponential mechanism, each with privacy budget $\varepsilon/k$, we will pick each coordinate $i$ such that $|\mu_i| \geq T$ with probability at least $1 - \beta/(4k)$. Taking a union bound over the success probabilities of the bounds, and the exponential mechanism rounds, we get that we will select $k$ coordinates above the threshold $T$ with probability $1 - \beta/2$ (if there are at least that many). Assuming we "give up" on potentially non-zero coordinates with mean magnitude lower than $T$, we get the following bound on the estimation error $\alpha$.

$$\sum_{i:|\mu_i| \leq T} \mu_i^2 \leq kT^2. \tag{35}$$

To get to an estimation error of at most $\alpha$, we need

$$\alpha^2 \geq kT^2 \geq 15k\frac{\sigma^2}{b} \geq 300k\sigma^2 \cdot \frac{k(\log d + \log(4k/\beta))}{n\varepsilon}. \tag{36}$$

This implies that for the support estimation part we will need

$$n = \Omega\left(\sigma^2 \frac{k^2(\log d + \log(k/\beta))}{\alpha^2 \varepsilon}\right) \tag{37}$$

samples. It remains to privately estimate the mean on the selected coordinates. Since we have already used up a super-linear (in $k$) number of samples, we can afford to use a naive estimator for the mean estimation part: we can invoke the histogram-based univariate estimator of Karwa and Vadhan [KV17] for each coordinate, and require accuracy of $\alpha/\sqrt{k}$ on each coordinate. For non-selected coordinates, we return $\hat{\mu}_i = 0$. In total, that would give us an $\ell_2$ guarantee of $\alpha$, as desired.

From [KV17, Theorem 1.1] we know that we need

$$n = \Omega\left(\frac{\sigma^2 k \log(2k/\beta)}{\alpha^2} + \frac{\sigma k^{1.5} \log(2k/\beta)}{\alpha\varepsilon} + \frac{k \log(R)}{\varepsilon}\right) \tag{38}$$

samples to estimate each coordinate up to under $\varepsilon/k$ DP with probability at least $1 - \beta/(2k)$. Using coordinate, we will need $\tilde{\Omega}(k^{1.5}/(\alpha^2\varepsilon))$ samples in total for the dense mean estimation part.

Combining the support estimation and the dense mean estimation components, we get the desired result.

$\square$

# G  Re-statement of sparse mean algorithm of [CWZ21]

In Section 3 and Section 4 we compare Algorithm 1 with [CWZ21, Algorithm 3.3] (referred to as CWZ algorithm hereafter). We focus solely on pure DP, and [CWZ21] state only an approximate DP version of the CWZ algorithm. To have a fair comparison, we make a (minor) modification of the CWZ algorithm to handle the $\varepsilon$-DP case. In short, we replace the advanced composition step in their analysis with a basic composition needed for the more stringent pure DP requirements.

More concretely, we keep [CWZ21, Algorithm 3.3] intact and only modify the peeling subroutine [CWZ21, Algorithm 3.2]. Given an a priori boun $R_\infty$ satisfying $\|\mu\|_\infty \leq R_\infty$, we replace the scale $R_\infty \cdot \frac{\sqrt{k \log(1/\delta)}}{\varepsilon}$ of the Laplace noise added on lines 3 and 7 of Algorithm 3.2 with $R_\infty \cdot \frac{k}{\varepsilon}$. The new privacy analysis differs from [CWZ21, Lemma 3.3] only in the composition steps.

# H  Meta-theorem for concentrated DP

An alternative formulation of differential privacy if that of zero-concentrated DP (zCDP) based on Rényi divergence. More formally, we have the following definitions.

**Definition H.1** (Rényi divergence). *Given distributions $P$ and $Q$ on a common sample space $\Omega$, the $\alpha$-Rényi divergence between $P$ and $Q$ is given by*

$$D_\alpha(P\|Q) = \frac{1}{\alpha - 1} \log\left(\mathbb{E}_{x \sim P}\left[\left(\frac{P(x)}{Q(x)}\right)^{\alpha-1}\right]\right). \tag{39}$$

**Definition H.2** (zCDP [BS16]). *Let $\mathcal{X}$ be a set and $\mathcal{X}^* = \{(X_1, \ldots, X_n) : n \in \mathbb{N}, X_i \in \mathcal{X}\}$ be all possible datasets over $\mathcal{X}$. For $\rho > 0$, a (randomized) map $M : \mathcal{X}^* \to O$ (where $O$ is an output set) is $\rho$-zCDP if for every $(X_1, \ldots, X_n), (X_1', \ldots, X_n') \in \mathcal{X}^*$ such that $X_i = X_i'$ except for a single index $i$ and for every $\alpha \in (1, \infty)$ it holds that $D_\alpha(M(X_1, \ldots, X_n)\|M(X_1', \ldots, X_n')) \leq \rho\alpha$.*

With this, we are ready to state the zCDP version of Theorem 2.1.

**Corollary H.3** (zCDP version of [Theorem 2.1]). *Let $M : \mathcal{X}^* \to \mathcal{O}$ be an $\rho$-zCDP map from datasets $\mathcal{X}^*$ to outputs $\mathcal{O}$. For every dataset $X_1, \ldots, X_n$, let $G_{X_1,\ldots,X_n} \subseteq \mathcal{O}$ be a set of* good *outputs. Suppose that $M(X_1, \ldots, X_n) \in G_{X_1,\ldots,X_n}$ with probability at least $1 - \beta$ for some $\beta = \beta(n)$. Then, for every $n \in \mathbb{N}$ and every $\delta > 0$, on $n$-element datasets $M$ is* robust *to adversarial corruption of any $\eta(n)$-fraction of inputs, where*

$$\eta(n) = \Omega \left( \min \left( \frac{\log 1/\beta}{\varepsilon(\rho,\delta) \cdot n}, \frac{\log 1/\delta}{\varepsilon(\rho,\delta) \cdot n + \log n} \right) \right) ,$$

*and $\varepsilon(\rho,\delta) = \rho + 2\sqrt{\rho \log(1/\delta)}$, meaning that for every $X_1, \ldots, X_n$ and $X'_1, \ldots, X'_n$ differing on only $\eta n$ elements, $M(X'_1, \ldots, X'_n) \in G_{X_1,\ldots,X_n}$ with probability at least $1 - \beta^{\Omega(1)}$.*

*Proof.* The result follows directly from the connection between zCDP and approximate differential privacy [BS16, Proposition 1.3]. □