# OpenReview forum: "Privacy Induces Robustness: Information-Computation Gaps and Sparse Mean Estimation"
_NeurIPS.cc/2022/Conference — NeurIPS 2022 Accept_

### Official Review · Reviewer_5phx · 2022-07-11

**Rating:** 6
**Confidence:** 2
**Soundness:** 3 good
**Presentation:** 4 excellent
**Contribution:** 3 good

**Summary:**

This paper works with differential privacy and a specific notion of robustness, namely the $\eta$-contamination. The authors show that DP mechanisms can be robust as well, and characterize a trade-off between computational efficiency and privacy for sparse mean estimation. Furthermore, more efficient mechanisms are proposed.

**Questions:**

While I understand the goal of sparse mean estimation as described in line 58-60, could the authors elaborate what is the application of sparse mean estimation? If such an application is empirically verifiable, the authors should experiment on it.

**Limitations:**

The authors have adequately addressed the limitations.

**Strengths And Weaknesses:**

The paper is clearly organized and sound as far as I can tell. The problems are important to work with and the improvement (from Table 1) is significant in the sense of poly time and robustness.

I think this paper is making a concrete contribution by proposing an actual mechanism that achieves the optimal tradeoff. However, my main concern is the motivation of the sparse mean estimation. See my questions below.

---

> ### Author Response · Authors · 2022-08-02
> **Response to Reviewer 5phx**
>
>
> We thank the reviewer for their constructive feedback.
>
> **Comment**: While I understand the goal of sparse mean estimation as described in line 58-60, could the authors elaborate what is the application of sparse mean estimation? If such an application is empirically verifiable, the authors should experiment on it.
>
> **Response**:
> Sparse estimation is a central topic in statistics [1] and sparse mean estimation is arguably one of the most fundamental problems in that field. While it is likely that there exist direct practical applications of private sparse mean estimation, this is not the main reason we focus on this problem. Mean estimation has proved to be an invaluable ''sandbox'' for studying statistical phenomena, paving the way for studying more sophisticated problems with broader practical applications. For example, in the case of robustness in high dimensions, the tools and insights developed for robust mean and covariance estimation [2] were key for a plethora of statistical tasks -- [3, 4, 5] among others. Here, we use mean estimation to study the interplay between differential privacy and sparsity in high dimensions.
>
> We thank the reviewer for bringing up this question; we will elaborate on the motivation behind sparse mean estimation in the updated version of the manuscript.
>
>
> [1] Hastie, Trevor, Robert Tibshirani, and Martin Wainwright. "Statistical learning with sparsity." Monographs on statistics and applied probability 143 (2015): 143.
>
> [2] Diakonikolas, Ilias, Gautam Kamath, Daniel Kane, Jerry Li, Ankur Moitra, and Alistair Stewart. "Robust estimators in high-dimensions without the computational intractability." SIAM Journal on Computing 48, no. 2 (2019): 742-864.
>
> [3] Diakonikolas, Ilias, Gautam Kamath, Daniel Kane, Jerry Li, Jacob Steinhardt, and Alistair Stewart. "Sever: A robust meta-algorithm for stochastic optimization." In International Conference on Machine Learning, pp. 1596-1606. PMLR, 2019.
>
> [4] Liu, Liu, Yanyao Shen, Tianyang Li, and Constantine Caramanis. "High dimensional robust sparse regression." In International Conference on Artificial Intelligence and Statistics, pp. 411-421. PMLR, 2020.
>
> [5] Chen, Sitan, Frederic Koehler, Ankur Moitra, and Morris Yau. "Kalman filtering with adversarial corruptions." In Proceedings of the 54th Annual ACM SIGACT Symposium on Theory of Computing, pp. 832-845. 2022.

---

### Official Review · Reviewer_Wizm · 2022-07-13

**Rating:** 7
**Confidence:** 3
**Soundness:** 3 good
**Presentation:** 3 good
**Contribution:** 3 good

**Summary:**

This paper initiates a study of how privacy, success probability and computation resources are trade-offed in DP sparse mean estimation problem. Firstly, the paper establishes a simple connection between robust and differentially-private algorithms: private mechanisms which perform well with very high probability are automatically robust in the sense that they retain accuracy even if a constant fraction of the samples they receive are adversarially corrupted. By this connection and assuming the Brennan Bresler secret-leakage planted clique conjecture, this paper has provided a lower bound for this problem given the computation constraints. Meanwhile, an efficient algorithm based on SOS is proposed in this work.

**Questions:**

1. line 221, $\Omega$ should be $O$.
2. line 306, $\alpha <<1$.

**Ethics Review Area:**

["I don’t know"]

**Strengths And Weaknesses:**

Strengths:
1. The paper has revealed a connection between DP algorithms and robust algorithms. I think it is a simple but interesting observation which may lead to more information-theoretic bounds for both fields.
2. The sparse mean estimation algorithm based on SOS seems to have made a sound algorithmic contribution to the community.

Weakness:
1. The constraints in the connection between DP and robustness is relatively stringent: it requires either the DP algorithm succeeding with a super high probability or the robust algorithm having theoretical guarantees for relatively low $\eta$.
2. The lower bound of sparse mean estimation depends on one conjecture, and meanwhile the upper bound only works for some range of parameters. There is still a non-trivial amount left before closing the gap for all parameter ranges.

---

> ### Author Response · Authors · 2022-08-02
> **Response to Reviewer Wizm**
>
> We thank the reviewer for their constructive feedback.
>
> **Comment**: The constraints in the connection between DP and robustness is relatively stringent: it requires either the DP algorithm succeeding with a super high probability or the robust algorithm having theoretical guarantees for relatively low $\eta$.
>
> **Response**: It is our goal to study the precise connection between private
> and robust algorithms, and the relationship we uncover is quantitatively tight.
> It is a feature, not a bug, that this reveals exactly what makes DP algorithms
> (automatically) robust -- that they succeed with high probability.
>
> ---
>
> **Comment**: The lower bound of sparse mean estimation depends on one conjecture.
>
> **Response**: It is standard to base computational complexity hardness results on well-studied conjectures (e.g. $P$ versus $NP$). We consider the planted clique conjecture, and in particular a minor variant studied by Brennan and Bresler. The planted clique conjecture is among the most well-studied in the field of average-case complexity, with a rich history dating back 30 years [1].
>
> ---
>
> **Comment**: The upper bound only works for some range of parameters. There is still a non-trivial amount left before closing the gap for all parameter ranges.
>
> **Response**:
> We agree with the reviewer that our method does not cover all range of parameters for the private sparse mean estimation. The restriction on the sparsity parameter $k$ in our method stems from a volume argument in the convex relaxation used to make the algorithm computationally efficient.  Nevertheless, we provide an algorithm that matches the conjectured trade-off in a polynomially-large range of sparsity regimes, as well as the practically relevant regimes of $\alpha,\epsilon > 1/ \mathrm{poly}\log d$. Finally, we would like to reiterate that we are not aware of any previous computationally efficient algorithms that match the tradeoff.
>
> ---
>
> **Comment**:
> line 221, $\Omega$ should be $O$; line 306, $\alpha << 1$.
>
> **Response**:
> Fixed in the updated version of the manuscript, we thank the reviewer for spotting these!
>
> [1] Jerrum, Mark. "Large cliques elude the Metropolis process." Random Structures & Algorithms 3, no. 4 (1992): 347-359.

---

### Official Review · Reviewer_4bHQ · 2022-08-07

**Rating:** 7
**Confidence:** 3
**Soundness:** 4 excellent
**Presentation:** 4 excellent
**Contribution:** 4 excellent

**Summary:**

Summary

The paper studies the connection between robustness and privacy. The paper begins by observing that the group privacy implies robustness, and group privacy follows by the standard (eps)-pure privacy as long as the failure probability of the (eps,0)-pure DP mechanism is small. A corresponding argument for the (eps,delta)-approximate DP also holds as long as delta is exponentially small.

This observation/meta-theorem allows the paper to borrow both computational and/or information-theoretic lower bounds from recent advances in high-dimensional robust statistics. The paper then derives various results for important statistical problems of the following flavor:  Consider the task A, then there is no (eps,0)-pure DP algorithm for task A that succeeds with probability delta, unless it either takes n >> n_0(eps,delta) samples (which is much larger than the sample complexity), or it does not run in polynomial time (under widely believed conjectures).

The main focus of the paper is on the problem of (Gaussian) sparse mean estimation. From [DKS17,BB20] it is known that the efficient algorithms for robust (Gaussian) sparse mean estimation requires quadratically more samples than the sample complexity. Combining this with the proposed meta theorem, the paper derives computational lower bounds for (eps,0)-pure DP mechanism that succeeds with high probability.

The main technical result of the paper is an efficient algorithm for private sparse mean estimation that matches the given computational lower bounds in the regime k \in [d^{0.4}, d^{0.5}]. The proposed algorithm is based on sum-of-squares and builds on the recent work of [HKM21].

**Questions:**

 The computational lower bounds of this paper hold only for those private algorithms that admit an exponentially small failure probability. I understand that there might be problems that admit private and polynomial-time algorithms with optimal sample complexity in constant-failure-probability regime but not in exponential-failure-probability regime (as shown by the paper, sparse mean estimation).
But even if a problem was hard in the constant failure probability regime (if there is such a problem), the meta-theorem of the paper will not be applicable. Do the authors have any comments on this limitation of their framework? Are there any candidates for private estimation tasks that might be hard even in the constant failure probability regime?



**Strengths And Weaknesses:**

# Strengths

I believe the paper makes important first steps in formalizing the connections between efficient algorithms for robustness and privacy.

The conceptual idea behind the meta theorem is simple and natural, which allows the paper to derive guarantees for many kinds of problems by borrowing results from algorithmic robust statistics.

I liked the paper, and thus I recommend acceptance.

# Weakness

+ The polynomial time algorithm for private sparse mean estimation works in a restricted parameter of regime.

---

> ### Author Response · Authors · 2022-08-09
> **Response to Reviewer 4bHQ**
>
> We thank the reviewer for their insightful comments.
>
> **Comment**:
> The polynomial time algorithm for private sparse mean estimation works in a restricted parameter of regime.
>
> **Response**:
> Please see response to reviewer Wizm (reproduced below for completeness).
> We agree with the reviewer that our method does not cover all range of parameters for the private sparse mean estimation. The restriction on the sparsity parameter $k$ in our method stems from a volume argument in the convex relaxation used to make the algorithm computationally efficient. Nevertheless, we provide an algorithm that matches the conjectured trade-off in a polynomially-large range of sparsity regimes, as well as the practically relevant regimes of $\alpha,\epsilon > 1/ \mathrm{poly}\log d$. Finally, we would like to reiterate that we are not aware of any previous computationally efficient algorithms that match the tradeoff.
>
> ---
>
> **Comment**:
> But even if a problem was hard in the constant failure probability regime (if there is such a problem), the meta-theorem of the paper will not be applicable. Do the authors have any comments on this limitation of their framework? Are there any candidates for private estimation tasks that might be hard even in the constant failure probability regime?
>
> **Response**:
> While we did not investigate any such problems in the paper (because we do not know of any), we disagree that the meta-theorem is strictly non-applicable in this case -- the picture is just a little more subtle. The meta-theorem can be applied to an $\epsilon$-DP algorithm which has constant failure probability to show that that algorithm is robust to adversarial corruption of an $\Omega(1/\epsilon n)$-fraction of its inputs -- that is, to $\Omega(1/\epsilon)$ inputs. If we had evidence of hardness (computational or information-theoretic) for some problem when $\Omega(1/\epsilon)$ of the inputs are corrupted, the meta-theorem then implies that this hypothesized $\epsilon$-DP algorithm shouldn't exist.
> Intuitively, the reason it's hard to find a problem like this which becomes computationally hard when $\Omega(1/\epsilon)$ of the inputs are corrupted is that in $n^{O(1/\epsilon)}$ time you can do brute-force search over all subsets of $1/\epsilon$ potentially-corrupted coordinates (hardness for $poly(n,1/\epsilon)$-time algorithms is still an interesting possibility in this context, which we haven't explored).
>
> One can, of course, derive lower bounds for private algorithms that do not go through a connection to robustness -- e.g. packing lower bounds in the case of pure DP, or fingerprinting codes in the case of approximate DP. One illustrative example is the case of univariate mean estimation for data in a region $[-R, R]$ under pure DP constraints. Simple packing arguments give us a $O(\log(R))$ term in the lower bound (regardless of failure probability) that is inherent to pure DP and does not appear neither in the standard non-private setting, nor in the robust setting.

---

### Meta-Review · Area_Chair_H7mD · 2022-08-30

**Recommendation:** Accept
**Confidence:** Certain

**Metareview:**

This work further explores the connection between outlier-robustness and differential privacy.
The first contribution of the paper is the following natural observation: group privacy implies robustness,
and group privacy follows by the standard pure privacy as long as the failure probability is small.
This allows the authors to derive lower bounds (information-computation tradeoffs and information-theoretic sample lower bounds)
for private estimation, as long as similar lower bounds are known in the robust setting. This includes Gaussian
sparse mean estimation (the focus of this work) and several others. The technical contribution of the paper is an efficient
algorithm for private sparse mean estimation that matches the lower bounds in a restricted parameter regime. This builds
on recent work by Hopkins et al. (STOC'22).  The reviewers agreed that this work is conceptually interesting and recommended
acceptance.


**Award:**

No

---

### Decision · Program_Chairs · 2022-09-14

Accept